# DIVERSITY MODELING FOR SEMANTIC SHIFT DETECTION

## ABSTRACT

Semantic shift detection faces a big challenge of modeling non-semantic feature diversity while suppressing generalization to unseen semantic shifts. Existing reconstruction-based approaches are either not constrained well to avoid over-generalization or not general enough to model diversity-agnostic in-distribution samples. Both may lead to feature confusion near the decision boundary and fail to identify various semantic shifts. In this work, we propose **Bi-directional Regularized Diversity Modulation (BiRDM)** to model restricted feature diversity for semantic shift detection so as to address the challenging issues in reconstruction-based detection methods. BiDRM modulates feature diversity by controlling spatial transformation with learnable dynamic modulation parameters in latent space. **Smoothness Regularization (SmoReg)** is introduced to avoid undesired generalization to semantic shift samples. Furthermore, **Batch Normalization Simulation (BNSim)** coordinating with auxiliary data is leveraged to separately transform different semantic distributions and push potential semantic shift samples away implicitly, making the feature more discriminative. Compared with previous works, BiRDM can successfully model diversity-agnostic non-semantic pattern while alleviating feature confusion in latent space. Experimental results demonstrate the effectiveness of our method.

## 1 INTRODUCTION

Deep neural networks (DNNs) have shown excellent performance in various computer vision applications (He et al., 2016; Cai & Vasconcelos, 2018). However, when facing samples beyond training distribution, DNNs will output wrong predictions with high confidence rather than decline to predict (Hein et al., 2019), due to the large network capacity and undesired generalization capability. This may cause potential security risk and limit the use of DNNs in open world. For this reason, deep learning community expects DNNs to reliably detect out-of-distribution (OOD) like humans. Recent studies (Yang et al., 2021) categorize generalized OOD detection into *covariate shift* (*e.g.* industrial defect, strange behavior, etc.) detection and *semantic shift* (*i.e.* different semantic class) detection, where this paper focuses on the latter[1].

Discriminative models (Hsu et al., 2020; Hendrycks et al., 2019a; Tack et al., 2020) detect OOD by explicitly learning discriminative features with the aid of true labels or pseudo OOD data, whose accessibility and quality may largely affect the detection performance. In contrast, reconstruction-based generative methods (Deng & Li, 2022; Floto et al., 2023; Liu et al., 2023b) focus on unsupervised in-distribution (ID) modeling, thus are more practical in real-world scenarios. The basic assumption is that models trained on ID data only will reconstruct any IDs well while assign OODs with high reconstruction error thus can detect them. In this paper, we follow this unsupervised setting and focus on *reconstruction-based* semantic shift detection.

In real-world scenarios, however, IDs with the same semantics usually exhibit high diversity (*i.e. non-semantic diversity*), while semantic shifted samples may have similar features with IDs (*i.e.* near-OOD). This brings a *trade-off dilemma* between modeling diverse IDs and suppressing the generalization to OODs for reconstruction-based OOD detection. To address this issue, some methods (Park et al., 2020; Liu et al., 2023a; Mou et al., 2023) propose to model ID diversity. Ideally,

---

[1]Without specification, OOD refers to semantic shift in the following text.

diverse IDs should be well-modeled by compact high-density region and the measure should be positively correlated to the severity of semantic shift, especially for samples near the boundary between ID and OOD. However, these methods are either not constrained well to avoid over-generalization to semantic shift (Park et al., 2020), or designed for *specific* non-semantic diversity patterns (*e.g.* geometrical anomaly (Liu et al., 2023a) and image-level corruption region (Mou et al., 2023)) which may fail in front of other types semantic shifts.

To address the above limitations, inspired by the illuminating discovery (Bengio et al., 2013) that meaningful non-semantic diversity can be controlled by spatial transformation in latent space, we design reconstruction-oriented **Bi-directional Regularized Diversity Modulation (BiRDM)** to modulate *diversity-agnostic* non-semantic pattern bi-directionally in feature space. BiRDM firstly removes non-semantic diversity by dynamic demodulation to capture semantic prototypes and then recovers non-semantic diversity by regularized modulation at reconstruction. To guarantee the effectiveness of this method, two challenging targets should be achieved: the bi-directional modulation cannot facilitate semantic shift samples to be well reconstructed, and the severity of semantic shift is measurable and discriminative in the latent representation space for ID (especially diverse IDs) and OODs (especially near-OOD).

For the first target, we assume a bounded representation space for diversity modeling and impose *smoothness regularization* to ensure smoothly changing modulation. In this way, relatively large diversity in untrained region (interval between training samples) can be avoided. And the diversity representation space could model non-semantic diversity well. For the latter one, we design *Batch Normalization simulation* branch to track prototypical feature for IDs and transform OODs separately in contrast to global affine transformation in standard Batch Normalization. This maintains more OOD information to be exposed to scoring function. On this basis, the additional optimization over auxiliary data with unified reconstruction task further drives potential OODs far away from the restricted non-semantic region and enhances discriminability to unseen samples. In this way, OODs could be distinguished from diverse IDs with respect to both feature distance measure and reconstruction error.

Our main contributions are as follows:

- We propose to model diversity-agnostic non-semantic feature with bi-directional regularized diversity modulation which is more feasible for semantic shift detection.

- We propose smoothness regularization to shape a well-constrained representation space for smoothly changing diversity modulation, which prevents semantic shift samples from being reconstructed with sharp diversity.

- We leverage Batch Normalization simulation with auxiliary data to allow separate transformations of demodulated features which enhance the discriminability between non-semantic samples and semantic samples.

## 2 RELATED WORK

**Out-of-distribution detection.** OOD detection approaches are expected to assign relatively high scores for OODs in unseen distribution. For this purpose, two compatible strategies are mainly studied, *i.e.* constructing decision boundary and modeling the potential IDs. Hsu et al. (2020); Zhu et al. (2022) learn classifier with ID labels, benefiting from implicit centralization hypothesis and more meaningful semantic features. Tack et al. (2020); Hendrycks et al. (2019a); Hein et al. (2019) additionally project real OODs or simulated pseudo OODs to a specified output for further confirming the boundary. However, the accessibility of labels and OOD data are doubtful. Even if they are available, ID labels or pseudo OODs may provide limited or biased information for properly separating OOD features from diverse IDs.

Reconstruction-based models (Akcay et al., 2018; Deng & Li, 2022; Liu et al., 2023b) typify the latter strategy, which are trained to reconstruct ID information and detect OODs with relatively high reconstruction error. As a combination of the two strategies, Oberdiek et al. (2022) takes reconstructed samples as pseudo OODs to learn the decision boundary for additional improvement. In this paper, we propose to extend reconstruction-based method by diversity modeling to enhance representation capacity for diverse IDs while inhibiting the generalization to unseen OODs.

**Feature transformation.** Feature transformation is widely used in controllable generation tasks. Researchers hope to edit semantic classes or non-semantic attributes along specified direction without interfering the others. AEs and GANs (Li et al., 2020; Ge et al., 2021) can take conditional inputs for independent and meaningful image editing. Style transfer methods (Dumoulin et al., 2017; Karras et al., 2019; 2020) introduce adaptive instance normalization to transform features in a specific direction to edit the styles and attributes. Feature transformation also has significant effect on latent space augmentation (Cheung & Yeung, 2021; Wang et al., 2022), which helps to enhance the generalization to non-semantic IDs and alleviate the limitation of image-level augmentation.

These works inspire us to perform spatial non-semantic diversity transformation in latent space, rather than directly generating the diverse embedding (*e.g.* residual connection) which is difficult to constrain. In addition, the dynamically generated transformation parameters is more feasible than static convolutional weights for decomposing and restoring the diversity of unseen distribution.

## 3 METHOD

In this section, we first review reconstruction-based semantic shift detection. Then, to address the problems of related methods, we present bi-directional regularized diversity modulation. After that, we expatiate on how to shape a smooth diversity representation space with smoothness regularization (SmoReg), and how to enhance the discriminability of unseen semantic shifts with Batch Normalization simulation (BNSim). Finally, we summarize the training and inference process.

### 3.1 REVIEW OF RECONSTRUCTION-BASED SEMANTIC SHIFT DETECTION

In reconstruction-based semantic shift detection framework, the autoencoder trained with only IDs is expected to produce high reconstruction error on OODs. Formally, for an input image $\boldsymbol{x}$ with encoded feature $\boldsymbol{f}$ and a decoder $g(\cdot)$, the reconstruction objective is $L = Dis\big(\boldsymbol{x}, g(\boldsymbol{f})\big)$, where $Dis(\cdot)$ is a distance/similarity function (*e.g.* $MSE$ for image-level reconstruction and $CosineSimilarity$ for feature-level reconstruction). To alleviate over-generalization to OOD, Gong et al. (2019) introduces a memory module $\phi(\cdot; \boldsymbol{z})$ on memory embedding $\boldsymbol{z}$ and improve the objective as follows:

$$L = Dis\big(\boldsymbol{x}, g(\phi(\boldsymbol{f}; \boldsymbol{z}))\big) + \gamma R(\phi), \tag{1}$$

where $R(\phi)$ is the prototypical constraint to further shrink model capacity. But this formulation makes non-typical IDs over-compressed to prototypes in memory and cannot be reconstructed well. To deal with the trade-off dilemma, a promising way is to perform well-constrained diversity modeling with $\psi^+(\cdot)$ and transformation function $\circ$ in the reconstruction process:

$$L = Dis\big(\boldsymbol{x}, g(\phi(\boldsymbol{f}; \boldsymbol{z})) \circ \psi^+(\boldsymbol{x})\big) + \gamma R(\phi, \psi^+). \tag{2}$$

Specifically, MNAD (Park et al., 2020) concatenates intermediate feature without extra constraint as the transformation function. DMAD (Liu et al., 2023a) uses sample space deformation fields on spatial sampling to model geometrical diversity, and imposes constraints on deformation strength and smoothness via $R(\psi^+)$. RGI (Mou et al., 2023) adopts learnable mask with constraint on total mask area to indicate corruption. However, these works are designed for specific diversity (*e.g.* geometrical deformation and corruption) and the transformation operators (*e.g.* spatial sampling and binary mask), so they are not general enough for diversity-agnostic semantic shift detection.

### 3.2 BI-DIRECTIONAL REGULARIZED DIVERSITY MODULATION

**The framework.** To address the problems in general unsupervised semantic shift detection, we propose **Bi-directional Regularized Diversity Modulation (BiRDM)** with the following objective:[2]:

$$L = Dis\big(\boldsymbol{f}, g(\phi(\boldsymbol{f} \circ \psi^-(\boldsymbol{f}); \boldsymbol{z}) \circ \psi^+(\boldsymbol{f}))\big) + \gamma R(\phi, \psi^+, \psi^-). \tag{3}$$

The main differences from previous works lie in two aspects: (1) Instead of adopting fixed-weight convolution for reconstruction in sample space, we model non-semantic diversity using *dynamically generated* transformation parameters in *feature space*, in order to capture more complex diversity patterns. (2) We present *bi-directional modulation*, where *demodulation* $\psi^-$ removes the

---

[2]In the following text, we write a shorthand form of the reconstruction distance as $Dis(\boldsymbol{f}, g(\boldsymbol{z}_{proto}^{comp}; \boldsymbol{\theta}^+))$ with compressed prototypical feature $\boldsymbol{z}_{proto}^{comp}$ and modulation parameter $\boldsymbol{\theta}^+$.

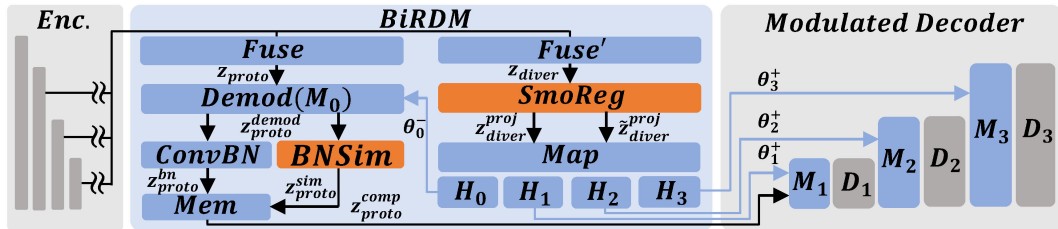

Figure 1: Overview of the proposed architecture. The fusion layer encodes features for prototype learning and diversity modeling respectively. Then regularized diversity feature from **SmoReg** (illustrated in Fig. 2) is used to generated dynamic modulation parameters. The demodulated feature is adapted by standard BN and **BNSim** (illustrated in Fig. 4) respectively, and compressed by memory network subsequently. Finally, the modulated decoder reconstructs diverse features.

non-semantic diversity to facilitate the consistency in demodulated prototypical feature, and *modulation* $\psi^+$ then recovers the diversity during the final decoding. In this way, our framework can optimize both non-semantic diversity modeling and prototypical learning.

Fig. 1 shows the BiRDM framework, in which some modules are explained and illustrated later. As shown, BiRDM first fuses feature $\boldsymbol{f}$ into *prototype feature* $\boldsymbol{z}_{proto}$ and *diversity feature* $\boldsymbol{z}_{diver}$ respectively. Since a semantic class may include spatially far fine-grained clusters (e.g. hairy cat and Sphynx in class "cat"), prototypical learning from $\boldsymbol{z}_{proto}$ is necessary to characterize discrete semantic factors, in conjunction with continuous non-semantic diversity modeling. Then, smoothness regularization is applied to get compact diversity feature $\boldsymbol{z}_{diver}^{proj}$, from which the modulation parameters are generated. After demodulation and memory compression, the feature $\boldsymbol{z}_{proto}^{comp}$ is reconstructed by modulated decoding.

**Diversity demodulation and modulation.** The parameters of demodulation ($\boldsymbol{M_0}$) and modulation ($\boldsymbol{M_1}, \ldots, \boldsymbol{M_3}$) are dynamically generated by a mapping network **Map** for distribution adaptation (Karras et al., 2019; 2020) and separate heads $\{\boldsymbol{H_0}, \ldots, \boldsymbol{H_3}\}$ for $N$-stages transformation operations. Denote the generation process with $h_{est}(\cdot)$, the demodulation parameters $\boldsymbol{\theta}^- = \{\boldsymbol{\theta}_{w,0}, \boldsymbol{\theta}_{b,0}\}$ and modulation parameters $\boldsymbol{\theta}^+ = \{\boldsymbol{\theta}_{w,1}, \boldsymbol{\theta}_{b,1}, \ldots, \boldsymbol{\theta}_{w,N}, \boldsymbol{\theta}_{b,N}\}$ are expressed as: $\{\boldsymbol{\theta}^-, \boldsymbol{\theta}^+\} = h_{est}(\boldsymbol{z}_{diver}^{proj})$. At the $n^{th}(n = 0, \ldots, N)$ stage, the demodulation/modulation performs affine transformation on any input feature $\boldsymbol{z}_n$:

$$\boldsymbol{z}_n^{mod/dmod} = \boldsymbol{\theta}_{w,n}(\boldsymbol{z}_n - \mu(\boldsymbol{z}_n))/\sigma(\boldsymbol{z}_n) + \boldsymbol{\theta}_{b,n}, \tag{4}$$

where $\mu(\cdot)$ and $\sigma(\cdot)$ are mean and standard deviation operator (Dumoulin et al., 2017; Karras et al., 2019; 2020) respectively. Though with the same formulation, demodulation and modulation have different objectives: to remove non-semantic diversity in prototypical learning, or recover diversity for the compressed prototypical feature $\boldsymbol{z}_{proto}^{comp}$.

**Modulation constraint.** Unconstrained modulation may make the model bypass memory module in prototypical learning, which causes shortcut learning and fails to detect semantic shift. Therefore, to ensure non-semantic diversity as small as possible under the condition that IDs can be well reconstructed in the bi-directional modulation process, we constrain the modulation parameters $\{\boldsymbol{\theta}^-, \boldsymbol{\theta}^+\}$ with the following loss:

$$L_{mod} = \sum_{n=0}^{N} \left\| \boldsymbol{\theta}_{w,n} - \mathbf{1} \right\|_2 + \left\| \boldsymbol{\theta}_{b,n} \right\|_2. \tag{5}$$

### 3.3 SMOOTHNESS REGULARIZED DIVERSITY MODELING

With the modulation constraint $L_{mod}$ restricting modulation amplitude, ID reconstruction quality may be degraded together with OODs. Another more important problem exists: there are *untrained regions* between training ID samples in the diversity feature space, which may generate large scale modulation parameters, leading to undesired OOD reconstruction (i.e. fail to detect). To address the above problems, we introduce sampling-based **Smoothness Regularization (SmoReg)** in diversity representation space as illustrated in Fig. 2. This regularization constrains the diversity modulation

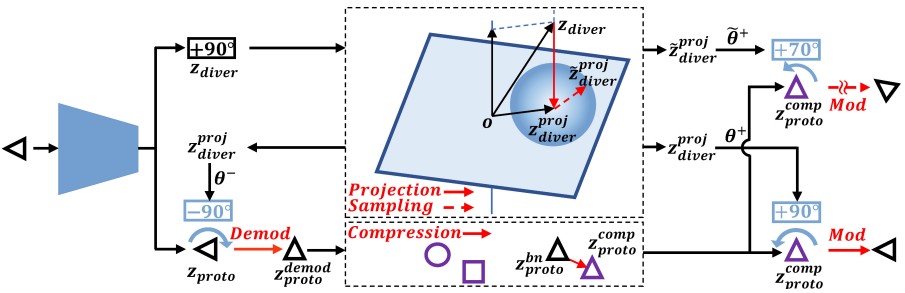

Figure 2: Illustration of SmoReg and prototypical learning in BiRDM. Assume rotation is diversity factor and shape is semantic prototype. ***upper***: $z_{diver}$ is projected to $z_{diver}^{proj}$ in diversity representation space and then sampled around. ***lower***: $z_{proto}^{comp}$ is addressed by the nearest neighbor of $z_{proto}^{bn}$.

to change smoothly along non-semantic ID manifold, by projecting diversity features to a more compact representation space where *IDs can be reconstructed with the modulation generated from neighboring sampled embeddings*. In this way, potentially high (low) reconstruction error for IDs (OODs) can be alleviated.

SmoReg firstly projects diversity feature $z_{diver}$ to a $D$-dimension bounded *diversity representation space* by orthogonal matrix $P \in \mathbb{R}^{C \times D}$ to capture more compact non-semantic factors:

$$z_{diver}^{proj} = min((P^T P)^{-1} P^T z_{diver}, a) \approx min(P^T z_{diver}, a), \qquad (6)$$

where $a$ is a constant restricting the maximal feature length. All training IDs are located in the diversity representation space as references. We perform sampling in the unoccupied regions between references to get more uniform feature distribution:

$$\tilde{z}_{diver}^{proj} \sim \mathcal{N}_{tr-inv}(z_{diver}^{proj}, 1, \tau_{tr}), \qquad (7)$$

where $\mathcal{N}_{tr-inv}(\cdot, \cdot, \tau_{tr})$ is an inverted $\tau_{tr}$-truncated Gaussian distribution, *i.e.* swapping the probabilities vertically within truncated region. Then, the sampled $\tilde{z}_{diver}^{proj}$ is used to get modulation parameters $\tilde{\theta}^+ = h_{est}(\tilde{z}_{diver}^{proj})$ for the corresponding reference. Finally, BiRDM additionally modulates compressed prototypical feature $z_{proto}^{comp}$ with modulation parameters $\tilde{\theta}^+$ during decoding. The smoothness objective can be expressed as constrained reconstruction loss:

$$L_{smo} = Dis(f, g(z_{proto}^{comp}; \tilde{\theta}^+)) + \|P^T P - I\|_2 + \|z_{diver} - P z_{diver}^{proj}\|_2, \qquad (8)$$

where the second term is orthogonal constraint and the last term restricts diversity feature $z_{diver}^{proj}$ being allocated in finite representation space. If $a$ is relatively small compared to sampling radius $\tau_{tr}$, minimizing $L_{smo}$ with aid of enough sampled $\tilde{z}_{diver}^{proj}$ (as explained in App. C) ensures smooth non-semantic diversity modeling (i.e. smoothly changing $\theta^+$) in the entire representation space.

**Remark.** SmoReg targets at smoothing the *unoccupied* regions in diversity representation space to suppress semantic shifts without affecting the real reconstruction of IDs. Optimizing smoothness objective also encourages ID projections to uniformly distribute in finite space to lessen the reconstruction confusion, which benefits BiRDM to mine

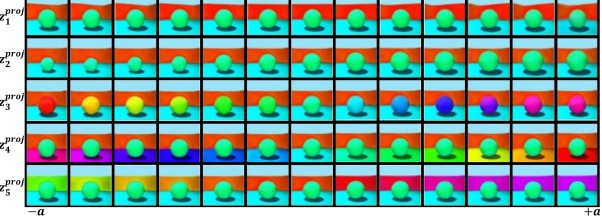

Figure 3: Disentanglement on 3DShapes.

*more* unoccupied regions to exclude OODs. In this way, the entire feature space can be properly modeled by smoothly changing non-semantic diversity. We also notice SmoReg associates with learning disentangled representation implicitly in Fig. 3, *i.e.* BiRDM could characterize non-semantic factors effectively.

### 3.4 SIMULATION OF BATCH NORMALIZATION

As shown in Fig. 1, the demodulated feature $z_{proto}^{demod}$ going through convolutional network and Batch Normalization (BN) becomes $z_{proto}^{bn} = ConvBN(z_{proto}^{demod})$. Then, the compressed prototypical fea-

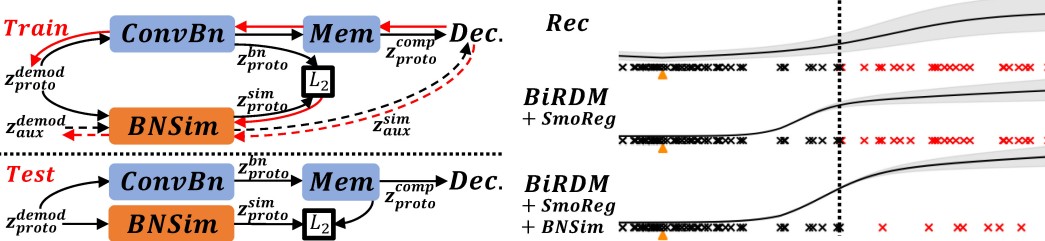

Figure 4: Detailed feature flow (black arrow) and gradient backpropagation (red arrow) in ConvBN and BN simulation. The dash line indicates auxiliary data.

Figure 5: Contribution of the proposed model to scoring function. Triangles are prototypes and gray areas indicate possible OOD score. The toy experiment is shown in Fig. 6.

ture is learned from the memory module as Liu et al. (2023a): $z_{proto}^{comp} = \arg\min_z \|z_{proto}^{bn} - z\|_2$, where the distance-based addressing mechanism benefits from feature centralization via standard BN. However, BN may incorrectly adapt OODs to training distribution with *global* affine transformation and decrease the feature discriminability (Zhu et al., 2022). Besides, training on finite ID data with prototypical consistency constraint (van den Oord et al., 2017), the sub-network can hardly keep necessary information for OOD detection.

To address this problem, we add a parallel branch of **Batch Normalization Simulation (BNSim)** implemented by $K$ *nonlinear-Layer* $h_{sim}(\cdot)$ without standard BN, as shown in Fig. 4. BNSim should have two capabilities: Firstly, it track standard BN outputs thus not disturb ID reconstruction, i.e. $z_{proto}^{sim} = h_{sim}([z_{proto}^{demod}]_{sg})$ is close to $z_{proto}^{bn}$ for all training samples. Note that we use stop-gradient operator $[\cdot]_{sg}$ to avoid updating related sub-network that forms a trivial solution.

Secondly, BNSim helps to enhance the discriminability between IDs and potential OODs. To this end, we further introduce auxiliary data $f_{aux}$ to train BNSim. The demodulated auxiliary features are transformed by BNSim to $z_{aux}^{sim} = h_{sim}(z_{aux}^{demod})$. Since SmoReg *constrains finite non-semantic diversity and the prototypical memory only characterize semantic factors of training IDs*, potential OODs in auxiliary data tend to distribute far from prototypes for better representation capacity. Without memory quantization (see App. A for more details), the auxiliary data should be well reconstructed. Therefore, the loss over BNSim is defined as:

$$L_{sim} = Dis\big(f_{aux}, g(z_{aux}^{sim}; \theta_{aux}^+)\big) + \big\|[z_{proto}^{bn}]_{sg} - z_{proto}^{sim}\big\|_2. \tag{9}$$

Since the highly nonlinear BNSim can better model warped transformation in contrast to global affine transformation of BN, optimizing the above objective will make ID and OOD more separable.

**Remark.** Different from OE (Hendrycks et al., 2019a), we optimize reconstruction error over auxiliary data, rather than explicitly learning discriminative feature to drive OOD to specified output. This formulation alleviates potential overfitting to specific $f_{aux}$, and is more feasible for contaminated data due to the unified optimization objective, *i.e.* BNSim does not need to exclude ID from auxiliary data. During inference, as shown in the lower part of Fig. 4, the output of BNSim preserves enough semantic information for OODs, so that the distance measure between BNSim output and prototypical memory can contribute to the scoring function for OOD detection.

### 3.5 TRAINING AND INFERENCE

**Training phase.** Following Liu et al. (2023a), we reconstruct training samples in feature space with VQ loss (van den Oord et al., 2017) to avoid computationally expensive image-level reconstruction:

$$L_{rec} = Dis\big(f, g(z_{proto}^{comp}; \theta^+)\big) + \big\|[z_{proto}^{bn}]_{sg} - z_{proto}^{comp}\big\|_2 + \beta\big\|z_{proto}^{bn} - [z_{proto}^{comp}]_{sg}\big\|_2. \tag{10}$$

According to the optimization framework of BiRDM in Eq. 3, we optimize the following overall loss with a hyperparameter $\gamma$ on the adversarial modulation constraint:

$$L_{all} = L_{rec} + \gamma L_{mod} + L_{smo} + L_{sim}. \tag{11}$$

**Inference phase.** Aside from reconstruction error, we also computes quantization error from BNSim, projection distance to restricted diversity representation space and modulation magnitude. The

Table 1: Semantic shift detection on CIFAR10 and FMNIST. * denotes reproduced results due to the absence of implementation, different settings or higher score than previously reported, their hyperparameters are searched on testing set (better than actual results). † denotes supervised methods.

(a) Per-class results on CIFAR10.

| Method | 0 | 1 | 2 | 3 | 4 | 5 | 6 | 7 | 8 | 9 | Avg. |
|---|---|---|---|---|---|---|---|---|---|---|---|
| GN | **93.5** | 60.8 | 59.1 | 58.2 | 72.4 | 62.2 | 88.6 | 56.0 | 76.0 | 68.1 | 69.5 |
| OCGAN | 75.7 | 53.1 | 64.0 | 62.0 | 72.3 | 62.0 | 72.3 | 57.5 | 82.0 | 55.4 | 65.6 |
| RD* | 89.4 | 89.4 | 76.8 | 65.6 | 80.6 | 78.0 | 82.2 | 85.6 | 91.4 | 84.8 | 82.4 |
| RD++* | 90.0 | 90.9 | 77.9 | 70.3 | 80.6 | 83.1 | 84.4 | 86.8 | 91.7 | 86.3 | 84.2 |
| Tilted* | 87.9 | 89.7 | 78.1 | 68.2 | 84.7 | 82.7 | 90.5 | 85.1 | 91.4 | 90.0 | 84.8 |
| VQ* | 90.7 | 93.1 | 81.1 | 66.9 | 88.2 | 78.2 | 89.9 | 89.5 | 93.4 | 88.8 | 86.0 |
| DMAD* | 89.1 | 91.9 | 77.5 | 76.7 | 85.9 | 81.6 | 89.2 | 89.7 | 91.8 | 85.2 | 85.9 |
| BiRDM | 92.0 | **96.7** | **85.9** | **81.3** | **89.9** | **89.9** | **95.2** | **92.9** | **96.1** | **93.7** | **91.3** |

(b) Per-class results on FashionMNIST.

| Method | 0 | 1 | 2 | 3 | 4 | 5 | 6 | 7 | 8 | 9 | Avg. |
|---|---|---|---|---|---|---|---|---|---|---|---|
| GN | 80.3 | 83.0 | 75.9 | 87.2 | 71.4 | 92.7 | 81.0 | 88.3 | 69.3 | 80.3 | 80.9 |
| AR | 92.7 | 99.3 | 89.1 | 93.6 | 90.8 | 93.1 | 85.0 | 98.4 | 97.8 | 98.4 | 93.9 |
| RD* | 92.5 | 99.5 | 91.4 | 94.5 | 92.0 | 95.9 | 81.3 | **99.5** | 94.9 | 98.5 | 94.0 |
| RD++* | 92.0 | 99.5 | 91.7 | 94.7 | 92.6 | 96.0 | 81.1 | 99.4 | 94.1 | 98.9 | 94.0 |
| Tilted* | 93.9 | 99.4 | 94.0 | 94.0 | 94.4 | 97.8 | 84.3 | 99.3 | 98.3 | 99.6 | 95.5 |
| VQ* | 94.3 | 99.5 | 93.7 | 94.7 | 94.9 | 96.5 | 84.1 | 99.3 | 97.3 | 99.4 | 95.4 |
| DMAD | 94.7 | 99.4 | 94.1 | 95.1 | 94.0 | 97.4 | 86.3 | 99.1 | 96.9 | **99.6** | 95.7 |
| BiRDM | **95.6** | **99.6** | **94.4** | **96.6** | **95.1** | **98.0** | 87.7 | 99.5 | 98.1 | 99.3 | **96.4** |

(c) Multi-classes results on unlabeled CIFAR10. OOD datasets are processed by fixed resize.

| ID | OOD | GOAD | Rot*† | GODIN*† | LMD | RD* | RD++*† | Tilted* | VQ* | DMAD* | BiRDM |
|---|---|---|---|---|---|---|---|---|---|---|---|
| C10 | LSUN (Fix) | 78.8 | 79.7 | 81.7 | - | 77.1 | 80.6 | 75.4 | 83.7 | 69.7 | **89.5** |
| | ImageNet (Fix) | 83.3 | 83.2 | 78.7 | - | 70.5 | 72.9 | 73.8 | 79.4 | 70.7 | **85.5** |
| | CIFAR100 | 77.2 | **79.1** | 76.0 | 60.7 | 65.9 | 64.5 | 74.4 | 74.9 | 69.5 | 77.8 |

final score used for semantic shift detection is computed as:

$$S = Dis\big(\boldsymbol{f}, g(\boldsymbol{z}_{proto}^{comp}; \boldsymbol{\theta}^{+})\big) + \alpha_1\big\|\boldsymbol{z}_{proto}^{sim} - \boldsymbol{z}_{proto}^{comp}\big\|_2 + \alpha_2\big\|\boldsymbol{z}_{diver} - \boldsymbol{P}\boldsymbol{z}_{diver}^{proj}\big\|_2 + \alpha_3 L_{mod}. \quad (12)$$

With diversity modeling, IDs will be perfectly reconstructed from memory (low term 1) and the demodulated feature after BNSim will be very similar to prototypes (low term 2). Meanwhile, OODs are projected far from prototypes (high term 2) and the reconstruction from discretely quantified ID embeddings has low quality (high term 1). For the last two terms, IDs are trained to allocate in the diversity representation space with compact modulation parameters, while OODs are more likely to distribute in the nullspace or out of the finite boundary with large modulation amplitude. Therefore, OODs will have high score $S$ and can be discriminated from low score IDs. For better understanding, Fig. 5 illustrates the contribution of the proposed model for detection scoring. Fig. 6 visualizes experimental results on toy data.

## 4 EXPERIMENT

### 4.1 IMPLEMENTATION DETAILS

We evaluate BiRDM on both one class and multiple classes semantic shift detection. BiRDM is based on reverse distillation (Deng & Li, 2022) and StyleConv (Karras et al., 2020). The encoder is ResNet18 for all reproduced approaches to ensure fairness. We employ 32 memory items with spherical initialization (see App. A for details) for uniformly addressing. We also use position embedding

Table 2: Ablation study and hyperparameters of BiRDM. Parameters in red are selected.

(a) Ablation study of proposed module and loss on one-class CIFAR10.

| | VQ*(86.0%) | | | | | | | | | | | | | | | | | |
|---|---|---|---|---|---|---|---|---|---|---|---|---|---|---|---|---|---|---|
| $\psi^+$ | ✓ | | | ✓ | ✓ | ✓ | ✓ | ✓ | ✓ | ✓ | ✓ | ✓ | ✓ | ✓ | ✓ | ✓ | ✓ | ✓ |
| $\psi^-$ | | ✓ | | ✓ | ✓ | ✓ | ✓ | ✓ | ✓ | ✓ | ✓ | ✓ | ✓ | ✓ | ✓ | | | ✓ |
| $L_{mod}$ | | | | | ✓ | | | | | ✓ | | ✓ | ✓ | ✓ | | | ✓ | ✓ |
| $L_{smo}$ | | | | | | ✓ | | | | | ✓ | ✓ | ✓ | | | ✓ | ✓ | ✓ |
| $\boldsymbol{P}$ | | | | | | | ✓ | | | | | ✓ | ✓ | | ✓ | ✓ | ✓ | ✓ |
| BN | | ✓ | | | | | | ✓ | ✓ | ✓ | ✓ | | | ✓ | ✓ | ✓ | ✓ | ✓ |
| $\Delta$AUC | 0.0 | 1.7 | 0.0 | 0.2 | 2.0 | 0.9 | 1.4 | 1.2 | 2.1 | 2.3 | 3.0 | 2.0 | 0.9 | 3.3 | 1.7 | 3.8 | 3.6 | 5.3 |

(b) Bi-directional modeling (best/mean/std)

| $\sharp\boldsymbol{z}$ | $\psi^+$-only | $\psi^-,\psi^+$ |
|---|---|---|
| 1 | 99.2/99.1/0.5 | **99.7/99.6/0.0** |
| 10 | 99.4/99.2/0.2 | **99.6/99.6/0.0** |
| 100 | **99.6**/99.3/0.2 | **99.6/99.5/0.1** |
| 200 | 99.7/99.0/0.7 | **99.8/99.5/0.2** |
| 1000 | **99.6**/99.1/0.5 | **99.6/99.5/0.0** |

(c) Memory quantity

| $\sharp\boldsymbol{z}$ | AUC |
|---|---|
| Const | 73.2 |
| $2^3$ | 73.1 |
| $2^4$ | 80.0 |
| $2^5$ | 81.3 |
| $2^6$ | **83.1** |

(d) Sampling quantity

| $\sharp\tilde{\boldsymbol{z}}_{diver}^{proj}$ | AUC |
|---|---|
| 0 | 75.2 |
| 1 | 76.0 |
| 15 | 80.5 |
| 25 | 81.3 |
| 40 | **81.6** |

(e) Sampling strategy

| $\tau_{tr}$ | $inv$ | AUC |
|---|---|---|
| $\boldsymbol{\theta}_{only}$ | | 75.2 |
| $\tilde{\boldsymbol{\theta}}_{only}$ | | 78.0 |
| -3dB | × | 79.3 |
| -10dB | ✓ | 78.8 |
| -3dB | ✓ | **81.3** |

to $z_{proto}^{comp}$ and a linear layer to adjust the decoded statistic to match modulation. Disjoint auxiliary data includes Tiny ImageNet and CIFAR100 is optimized every 5 batches for efficiency. We set $\beta = 0.25$, $\gamma \in \{0.1, 1\}$ depending on how diverse the training distribution is, and pseudo positives for validation can be obtained by rotation, blurring, etc. The optimizer is AdamW (Loshchilov & Hutter, 2017a) with CosineAnnealingLR (Loshchilov & Hutter, 2017b) whose learning rate is 3e-3 initially. We adopt 200 epochs for CIFAR10 and 40 for FashionMNIST on 1 GPU with the batch size of 8. We then evaluate area under receiver operating characteristics curve.

## 4.2 Main results

Tab. 1 provides comparative results with SOTA reconstruction-based models, where CIFAR10 is widely compared and not simpler than large-scale dataset as Yang et al. (2022) discussed (see App. C for more quantitative and qualitative results). All dataset are processed by fixed resize operation (Tack et al., 2020) which is more difficult for detection. GN (Akcay et al., 2018) and OCGAN (Perera et al., 2019) are GAN-based reconstruction works, and LMD Liu et al. (2023b) use Diffusion instead. RD (Deng & Li, 2022) is the baseline of the second group due to the *same* architecture. RD++ (Tien et al., 2023) optimizes OT for compactness (we create large-batch supervised variant to avoid collapse). Tilted denotes tilted-Gaussian (Floto et al., 2023) extension of RD. VQ denotes discretely quantified RD (Liu et al., 2023a), like prototype learning in DMAD and BiRDM. Note that reconstructing auxiliary data causing lower performance for these methods due to the undesired generalization. The other results are borrowed from Ye et al. (2022) and Tack et al. (2020).

Tab. 1(a) shows BiRDM significantly outperforms comparative reconstruction-based works on CIFAR10 whose non-sematic diversity is rich enough and hard to capture. Although most FashionMNIST data can be modeled by geometrical diversity only, Tab. 1(b) shows diversity-agnostic BiRDM outperforms previous works and geometric-specified diversity modeling in DMAD, demonstrating BiRDM can capture these geometrical diversity implicitly (see App. B and C for more discussion about DMAD and FashionMNIST). BiRDM also can discover the discrete semantic factors for multi-class ID and achieve great improvement on the detection performance as shown in Tab. 1(c).

## 4.3 Ablation study and analysis

**Bi-directional modulation.** We evaluate the capability of removing diversity by dynamically generated transformation parameter instead of static CNN on CIFAR10 and Ped2. As shown in Tabs. 2(a) and 2(b), bi-directional dynamic modeling is more stable to eliminate diversity and has higher performance than static CNN. We use average quantization distance (AQD) between demodulated

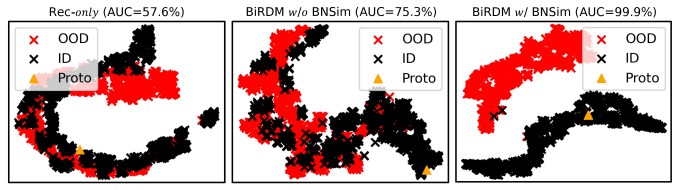 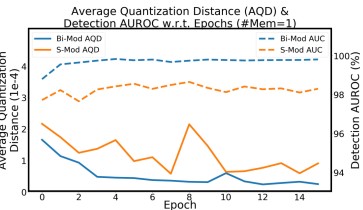

Figure 6: T-SNE visualization on dSprite. Semantic and co-variate shift are about shapes and position respectively. AUC is calculated by term 2 in Eq. 12. See App. C for more details.

Figure 7: Illustration of average quantization distance and detection result w.r.t. training epoch in Ped2.

feature and memory embedding as measure of non-semantic diversity. Fig. 7 shows our framework always has lower and smoother AQD, as well as faster convergence than the original one.

**Proposed components.** Tab. 2(a) shows the ablation of proposed components. Feature space modulation $\psi^+(\cdot)$ restores non-semantic diversity from prototypes to reconstruct high-quality features and the reverse process $\psi^-(\cdot)$ benefits prototypical learning to capture invariant prototypes. More importantly, SmoReg $L_{smo}$ creates a smooth representation space for non-semantic diversity which suppresses the transmission of semantic information. There is extra gain by using projection $\boldsymbol{P}$ to compress redundant diversity information in high dimensionality and perform sampling efficiently. The modulation constraint $L_{mod}$ further limits the intra-class scatter of modulated representation and improves the distribution compactness. Finally, BNSim maintains more discriminative features and makes the distance between demodulated feature and memory embedding more reliable.

**SmoReg and BNSim.** Comparing column 5 with 6-8 in Tab. 2(a), although $L_{mod}$, $L_{smo}$ and $\boldsymbol{P}$ alleviate the undesired OOD reconstruction, they force encoder to discard discriminative information about unseen OODs for further compression. Fortunately, the adverse effect is offset by BNSim in column 10-12. However, comparing column 5 with 9, we find BNSim alone brings limited improvement without SmoReg and the combinations always outperform others. Visualization on dSprites also shows the separation qualitatively in Fig. 6. These results verify the assumption in Sec. 3.4 that SmoReg and BNSim are interdependent.

**Hyperparameters.** As shown in Tab. 2(c), we use 32 memory items in all experiments for simplicity. Const input (static learnable embedding $\boldsymbol{z} \in \mathbb{R}^{2 \times 2 \times C}$ (Karras et al., 2019; 2020)) is also insufficient, though it is better than position-agnostic VQ-Layer with the same memory quantity due to the implicit positional information.

Tab. 2(d) confirms the efficacy of smoothness sampling. As the sampling times increase, diversity representation space is more completely covered and semantic shift samples are better pushed away from the distribution. Limited by the computational cost, we only perform $25\times$ samplings.

Tab. 2(e) shows there may not be enough pressure for using real samples $\boldsymbol{\theta}^+$ only to cover gaps between untrained region smoothly due to the absence of sampling-based regularization or reverse operation $w/o\text{-}inv$. And using sampled embedding $\tilde{\boldsymbol{\theta}}^+$ only may be biased without enough sampling quantity. Truncation technique $\tau_{tr}$ further alleviates the problem that embedding drawn from the low-density region of sampling distribution is hard to reconstruct the corresponding reference.

## 5 CONCLUSION

In this paper, we propose a diversity-agnostic modeling framework, BiRDM, for reconstruction-based semantic shift detection. BiRDM enhances the representation capacity to non-semantic patterns without semantic shifts. To this end, we apply smoothness regularization to perform smoothly changing modulation for ID diversity in latent space, which essentially exclude potential OOD reconstruction with large diversity. Batch Normalization simulation further separates and pushes unseen semantic feature away without disturbing ID representation and enhances the feature discriminability. In semantic shift detection experiments, BiRDM shows effectiveness and outperforms existing diversity-specific modeling approaches. In future work, we will enhance the disentanglement of BiRDM, so that more complete non-semantic factors could be excluded from semantic shift detection during inference.

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

## A More implementation details

**Detailed architectures.** The architectures of all approaches we reproduced follow the settings of RD (Deng & Li, 2022), which contain three stages of different scales. The fusion layers $Fuse$ and $Fuse'$ exactly match the one-class bottleneck embedding, whose dimensions are $\{64, 128, 256\} \rightarrow \{256, 256, 256\} \rightarrow \{256\}$. The only difference is that *prototype branch* outputs by Batch Normalization to adapt to similarity-based memory mechanism, while *diversity branch* adopts convolutional layer instead. Besides, the position embedding and scale adapter are implemented by $1 \times 1$ *Coord-Conv* (Liu et al., 2018). We use scaled ResNet decoder to adapt to modulation scales, *i.e.* perform scale normalization implicitly basing on estimated statistics of features (Karras et al., 2020). Then extra $N$ output heads without scaling are employed to reconstruct original features.

**Detailed implementation of BNSim.** As shown in the main paper Fig. 4, there are three different inputs obtained by demodulation, namely training data $\boldsymbol{z}_{proto}^{demod}$, auxiliary data $\boldsymbol{z}_{aux}^{demod}$ used in training phase and the testing data $\boldsymbol{z}_{proto}^{demod}$. For each kind of input, we have a standard ConvBN branch (upper) and BNSim branch without BN operation (lower).

- **Training data:** The standard ConvBN branch is trained with $VQ$ loss (van den Oord et al., 2017) and reconstruction loss (Deng & Li, 2022), whose gradient flow is same as VQ-VAE. Meanwhile, BNSim parallelly simulates the output of standard BN in the upper branch, constraining by $MSE$ loss (shown as the square mark $L_2$). Note that this part of gradient flow is only applied in BNSim branch without updating standard ConvBN branch or the previous network.

- **Auxiliary data:** The standard BN branch is omitted and the output of BNSim is used for decoding in case of overwriting the prototypical memory with auxiliary data. In this stage, we do not stop the gradient flow from final reconstruction loss, which forces the encoder to project potential OODs far away from high-density region to optimize reconstruction loss over auxiliary data. Because the region near demodulated prototypical feature and diversity representation space is fully occupied by non-semantic IDs and hard to reconstruct auxiliary data.

- **Testing data:** In inference phase, the standard ConvBN branch is used for reconstruction and obtaining reconstruction error. Because OOD features are expected to be discarded by memory module, their reconstruction error will be high. Furthermore, we use the output of BNSim to compute a distance-based metric (term 2 in Eq. 12, shown as the square mark $L_2$). The highly nonlinear BNSim can model warped transformation in contrast to the global affine transformation of standard BN branch, which make the separation of ID and OOD possible. In other words, discriminative OOD features could be kept in BNSim and significantly different from the prototypical memory.

**Reverse operation in sampling.** We show the result without reverse operation *-inv* in Tab. 2(e) in the main paper, where the performance decreases (AUC: $81.3\% \rightarrow 79.3\%$) with poor distribution uniformity. Actually, the reverse operation encourages stronger exploration to hard samples, which helps to aggressively shape the potential boundaries. This is similar to a relatively high $\alpha = \beta$ in $Beta(\alpha, \beta)$ distribution for Mixup (Zhang et al., 2018) to perform augmentation (another alternative is the Tilted Gaussian distribution). With the reconstruction of additional reference samples, the overall distribution is more uniform and the bias caused by insufficient samples will be alleviated.

**Spherical memory initialization.** As shown in Fig. 8, the uniform memory initialization (van den Oord et al., 2017) may cause BN centralized prototypes be addressed unevenly and updated unstablely. And different prototypes belonging to the same semantic class cannot share the diversity representation space. As a solution, we perform spherical memory initialization which normalizes the isotropic Gaussian distribution to make the memory items distributed on a hypersphere thus being addressed more uniformly:

$$\boldsymbol{z} = r \frac{\boldsymbol{z}}{\|\boldsymbol{z}\|_2}, \ \ \boldsymbol{z} \sim \mathcal{N}(\boldsymbol{0}, \boldsymbol{1}). \tag{13}$$

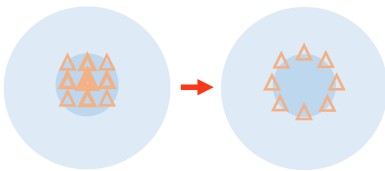

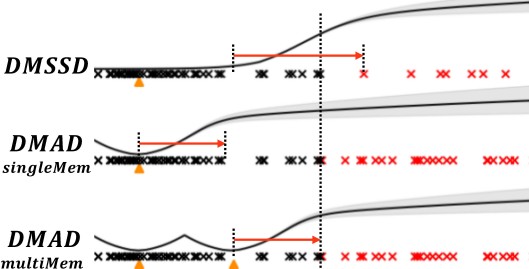

Figure 8: Illustration of spherical memory initialization. The density is indicated by brightness, prototypes by triangles and addressing probability by thickness. *left:* Uniform initialization makes the addressing probability not uniform. *right:* Isotropic spherical initialization allows the memory equally addressed.

Figure 9: Comparing BiRDM with DMAD. The red arrow indicates potential positively correlated region.

## B COMPARISON WITH DIVERSITY MEASURABLE METHOD

**Different target scenarios.** Despite the promising framework in DMAD, the precondition about positive correlated measurement may be too strong in the face of diversity-agnostic semantic shift. It is difficult to find a diversity-measurable implementation beyond geometrical diversity to meet the condition of positive correlation, *i.e.* DMAD with deformation estimation is diversity-specific and works well on semantic shift only if they *mainly involve geometrical diversity*. However, diversity of natural images is determined by extremely complex generative factors, and we cannot find such a group of diversity like diformation to model. As a result, DMAD outperforms VQ on FashionMNIST and MNIST, but fails in the more diverse CIFAR dataset (one-class: VQRD $86.0\%$ vs DMAD $85.9\%$; multi-class: VQRD $83.7\%/79.4\%/74.9\%$ vs DMAD $69.7\%/70.7\%/69.5\%$).

BiRDM provides a relaxed version of the three conditions described in DMAD and shapes the latent space in an essentially different way. SmoReg constrains a more linear smooth space for this purpose and expand the discrete prototypes to non-semantic representation space for diversity modeling. It is naturally to assume that the smooth margin at the boundary of high-density region ensures the separation of semantic shifted features and OODs will incur significant spatial difference. This diversity-agnostic modeling capacity is more important in generalized semantic shift detection for natural images.

**Different learning frameworks.** In DMAD, the revere process of deformation estimation in PPDM version is proposed to restore original information as cycle-consistency constraint. However, prototype learning is not well-separated from diversity modeling process. As a result, the redundant diversity information attached to semantic features may lead to unstable prototypical learning. On the contrary, BiRDM adopts bi-directional modulation framework where the demodulation is optimized by prototype learning almost being separated from reconstruction to remove non-semantic diversity and obtain demodulated prototypical features. Thus the prototype learning will be more robust.

**Different measurement.** As shown in Fig. 9, DMAD provide reliable anomaly score for relatively typical diversity, *i.e.* projection close to prototypical appearance, but fails to cover the entire non-semantic space with diversity modeling (middle subfigure). For highly diverse ID class, DMAD can cover ID region by using more prototypes but the high-score region may still distribute among IDs (bottom subfigure). Unlike DMAD, BiRDM takes each training sample as reference to construct smooth ID region with low OOD score. Therefore, the gaps between ID reference are much more slight and the discriminative region is located at the boundary between diverse IDs and OODs (upper subfigure).

**DMAD is a special case of BiRDM.** Actually, surface defects and behavioral anomaly could be *mainly* characterized by the deformation amplitude. The geometrical-specific diversity modeling approaches satisfy the positive correlation (Liu et al., 2023a) between the measurement and the severity of shift. In this case, the smoothness constraint penalizes spatial gradient of geometrical transformation, which prevents the excessively warped deformation from reconstructing anomalies. In another

Table 3: Ablation study on the number of modulation stages.

| ♯ modulation stages | 0 | 1 | 2 | 3 | 4 |
|---|---|---|---|---|---|
| AUC | 57.6 | 97.5 | 99.0 | 99.9 | **100.0** |

word, relatively small deformation naturally occupies a compact and smooth representation space in image-level and has effect similar to SmoReg locally. This is the reason why the implementation in DMAD is specifically designed, *i.e.* there is inductive bias for designing geometrical-specific structure without accessing or transforming deep features.

## C MORE EXPERIMENTAL DETAILS

### C.1 DATASETS

We evaluate our method on CIFAR, FashionMNIST and ImageNet, with the fixed-version OOD dataset (Tack et al., 2020).

**CIFAR10.** CIFAR10 (Krizhevsky & Hinton, 2009) is 32×32 resolution natural image dataset with 10 classes (*e.g.* airplane, cat, bird, etc), containing 50,000 training data and 10,000 testing data. Though small and simple, CIFAR10 is more challenging than ImageNet surprisingly (Yang et al., 2022) due to the diverse appearance and low resolution.

**FashionMNIST.** FashionMNIST (Xiao et al., 2017) includes 10-class fashion products (*e.g.* dress, coat, bag, etc). They are in $28 \times 28$ resolution and contain 60,000 training data and 10,000 testing data. Although this dataset could be used for semantic shift detection, some classes, *e.g.* T-shirt/top, pullover, coat and shirt, are more similar to covariate shift in fact. It may be difficulty for semantic shift detection as they have similar semantic features.

**ImageNet30.** ImageNet30 (Hendrycks et al., 2019b) is a subset of ImageNet, which contains 39,000 training images within 30 semantic categories and 3,000 images for ID testing data.

**OOD dataset.** LSUN and ImageNet excluding the overlapped classes with the training CIFAR10 are used as OOD dataset. And CUB200 (Wah et al., 2011), Places365 (Zhou et al., 2018) and Caltech256 (Griffin et al., 2007) are used for ImageNet30. As in Tack et al. (2020), we apply fixed resize operation to get hard OODs, otherwise they are easily recognized due to resizing noise.

**Auxiliary dataset.** We use CIFAR100 and Tiny ImageNet without overlapping with the test classes as the auxiliary dataset in all experiments, *i.e.* we use TinyImageNet if OOD dataset includes CIFAR100 to ensure fairness. Note that it is not necessary to make the auxiliary dataset to simulate near-OOD in test dataset or manually exclude IDs, as our purpose is to maintain necessary discriminative information with a unified reconstruction objective.

**Ped2.** Ped2 (Mahadevan et al., 2010) is a fixed-view surveillance videos for anomaly (including both semantic and covarate shift) detection, include driving, cycling, etc. as anomalies.

**Toy dataset.** We show the effect of BiRDM by toy experiments on dSperite (Higgins et al., 2017) and 3DShape (Kim & Mnih, 2018), which are widely used in representation learning due to the controllable generative factors.

### C.2 PARAMETER SETTING

As discussed in Sec. 3.4 and 4.1 in the main paper, the hyperparameters are selected according to previous works or validation set. Here we explain the effect of these parameters intuitively.

**Number of modulation stages:** As shown in Tab. 3, we test our model on dSprite and find BiRDM is not sensitive to the number of modulation stages as long as it is not too limited.

Table 4: Ablation study on the weight $\gamma$ of modulation constraint $L_{mod}$.

| $\gamma$ | Baseline (VQ*) | 0.1 | 0.25 | 0.5 | 0.75 | 1 | {0.1, 1} |
|---|---|---|---|---|---|---|---|
| CIFAR10-OC | 86.0 | 90.4 | 90.7 | 90.7 | 90.8 | 90.9 | **91.3** |
| FMNIST-OC | 95.4 | **96.4** | 96.3 | **96.4** | 96.3 | 96.2 | **96.4** |

Table 5: Multi-classes results on unlabeled large-scale ImageNet. * denotes reproduced results. † denotes supervised methods. [] denotes method could be improved by "flipping labels".

| ID | | | | ImageNet30 | | | | | ImageNet100 | | | |
|---|---|---|---|---|---|---|---|---|---|---|---|---|
| OOD | CUB | Places | Caltech | Dogs | Pets | Flower | Food | DTD | Naturalist | SUN | Places | DTD |
| RD* | 62.6 | 64.3 | 54.6 | 67.2 | 68.0 | 87.4 | 65.4 | 81.5 | 85.0 | 60.3 | 56.8 | 81.9 |
| RD++*† | 57.4 | 61.3 | 50.6 | 70.3 | 73.0 | 91.0 | 66.6 | 85.5 | 83.5 | [49.9] | 52.8 | 80.4 |
| Tilted* | 66.9 | 61.8 | 58.0 | 78.8 | 81.4 | 90.4 | 67.8 | 79.6 | 73.4 | 76.1 | 72.3 | 76.4 |
| VQ* | 67.0 | **66.1** | 66.4 | 80.3 | 81.9 | 87.7 | 67.9 | 82.5 | 60.7 | 64.5 | 58.6 | 76.5 |
| DMAD* | 67.7 | 63.7 | 62.2 | 74.2 | 76.3 | 87.8 | 79.3 | 73.9 | **90.2** | 64.9 | 58.7 | 81.7 |
| BiRDM | **82.0** | 65.5 | **70.1** | **84.8** | **87.2** | **93.7** | **86.0** | **89.9** | 86.8 | **85.6** | **80.7** | **85.9** |

$\gamma$: We only make adjustment via $\gamma$ for the modulation constraint $L_{mod}$ during training, because it is the only term that directly restricts diversity and fights against reconstruction target as discussed in Sec. 3.3. Therefore, we use relatively large (1) $\gamma$ for reconstructing IDs with limited diversity and small one (0.1) for allowing to capture more diverse variations in a single non-semantic latent variable. Nevertheless, we find $\gamma$ is not sensitive as shown in Tab. 4.

$\alpha_1$ & $\alpha_2$: The quantization error weighted by $\alpha_1$ and the projection distance weighted by $\alpha_2$ represent the severity of semantic shift from the perspective of semantic factors and non-semantic diversity factors respectively. For typical semantic shift detection, the quantization error is sufficiently discriminative that $\alpha_1$ could achieve to 1. Besides, when BiRDM is used to detect covariate shift (with similar semantics but diverse non-semantic information), $\alpha_1$ and $\alpha_2$ may be reduced to 0, to avoid confusion between OODs and diverse IDs.

$\alpha_3$: The modulation constraint $L_{mod}$ could make modulation after imposing SomReg more compact. So that this term will be discriminative if the sampling in SmoReg is insufficient to cover entire diversity space for natural images or OODs are unexpectedly included between extremely diverse ID projections as discussed later. That is $L_{mod}$ essentially identify data beyond training distribution with covariate shift or near-OOD.

## C.3 MORE EXPERIMENT

**Large-scale dataset.** Although CIFAR10 is hard enough as discussed in Yang et al. (2022) with sufficient non-sematic diversity shift, for example, the class "bird" contains different species (bird / ostrich / peacock), size (distant / close-shot image), background (sky / grass / tree, ...), color, posture, etc. We additionally include a large-scale benchmark (unlabeled ImageNet30 vs CUB / Places / Caltech / Dogs / Pets / Flowers / Food / DTD) (we do not mix OOD datasets or perform balanced sampling as in previous works to ensure the fairness) and a more challenging one (unlabeled ImageNet100 vs iNaturalist / SUN / Places / DTD) in Tab. 5. Furthermore, all resize operation used in is the fixed one as described in Tack et al. (2020) and we use grid search for model selection. The experimental results basically support our observation in main paper Tab. 1.

**Model architecture and capacity.** As reported in previous works, Transformer features are discriminative for classification-based approaches (Fort et al., 2021). We use features from vision Transformer family (PVTv2-B0, PVTv2-B1, PVTv2-B2 (Wang et al., 2021), where PVTv2-B1 is a pyramid vision transformer with similar parameters like ResNet18) and ResNet family (ResNet50, WideResNet50) as our distillation target. Note that the decoder is still ResNet rather than ViT-based model, because LayerNormalization may destroy the style-based modulation. Interestingly, it is

Table 6: Ablation study of model architecture on multi-class and one-class CIFAR10.

| Model | Encoder | Decoder | LSUN (Fix) | ImageNet (Fix) |
|---|---|---|---|---|
| BiRDM | PVT-V2-B0 | ResNet18 | ↓ 8.8 | ↓ 9.5 |
| BiRDM | PVT-V2-B1 | ResNet18 | ↓ 0.6 | ↑ 2.8 |
| BiRDM | PVT-V2-B2 | ResNet18 | ↓ 4.3 | ↓ 4.9 |
| VQ* | PVT-V2-B1 | ResNet18 | ↓ 0.1 | ↑ 3.5 |
| BiRDM | ResNet50 | ResNet50 | ↓ 3.1 | ↑ 2.1 |
| VQ* | ResNet50 | ResNet50 | ↓ 2.0 | ↑ 2.9 |
| BiRDM | WResNet50 | WResNet50 | CIFAR10-OC: ↓ 0.2 | |
| BiRDM | PVT-V2-B1 | ResNet18 | CIFAR10-OC: ↓ 3.9 | |

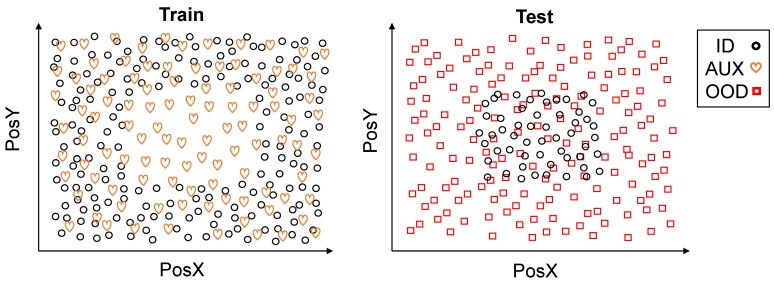

Figure 10: Data splits of dSprites used in Fig. 6.

hard to reproduce the same observations as Fort et al. (2021) in reverse distillation (reconstruction-based) methods as shown in Tab. 6. Larger capacity in a single model family does not mean better OOD detection performance, since only data from the same domain benefits from larger pre-trained models.

According to these results, we have following conjectures: 1. Using more diverse data to train a model will benefit OOD detection by maintaining more unseen semantic features (see discussion about BNSim and auxiliary data in main paper Secs. 3.4 and 4.3); 2. Model architecture has influence on reconstruction-based OOD detection, but larger model capacity does not always bring better performance. (The performance gap in anomaly detection caused by different model architectures is also reported by Heckler et al. (2023)); 3. We need a network with comparable scale to perform distillation/reconstruction, rather than using small networks for richer features; 4. Transformer features are effective in classification-based methods but may fail in reconstruction-based ones; 5. Pre-trained networks with larger scale always perform better on OOD dataset within the same domain; 6. Transformer backbone cooperating with CNN decoder may lead to training instability even with comparable scale (PVT-V2-B1+ResNet18: $std = 2.2AUC\%$).

**Toy experiment in Figure 6.** Inspired by Montero et al. (2022), we create a toy experiment to visualize how BiRDM shapes the latent space. Based on dSprites dataset, the data splits are shown in Fig. 10: 1. Training data is composed of ellipses located away from the centric region, *i.e.* coordinates (PosX and PosY) to center $> \tau$; 2. Auxiliary data contains hearts at all locations; 3. Testing data contains ellipses at the centric region (IDs), and squares (OODs) at all locations; The three BiRDM models are: BiRDM without SmoReg and BNSim; BiRDM with SmoReg only; full-components BiRDM.

**Demodulation with different memory quantity.** We also show average quantization distance for different memory quantity setting in Fig. 11, which supports the ablation study in main paper Sec. 4.3. Although more memory capacity means more stable convergence, the bi-directional modeling is always better than the original one.

**Computational cost.** We test models on Intel 7700k and Nvidia GTX1080ti. The model parameters and speeds are shown in Tab. 7. BiRDM has similar parameters as the compared approaches,

Table 7: Computational cost on CIFAR10.

| Model | RD* | RD++* | Tilted* | VQ* | DMAD* | BiRDM |
|---|---|---|---|---|---|---|
| Params (M) | 24 | 25 | 22 | 24 | 21 | 25 |
| Training Speed (min/epoch) | 5 | 100 | 6 | 5 | 9 | 12 |
| Testing Speed (FPS) | 129 | 117 | 131 | 137 | 76 | 72 |

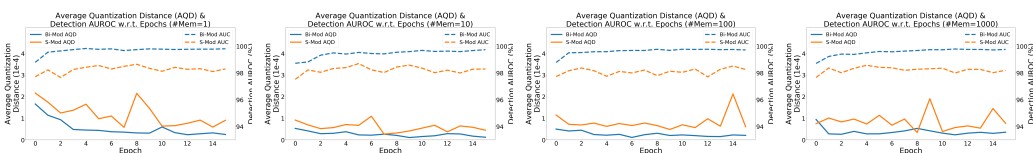

Figure 11: Illustration of average quantization distance and detection result w.r.t. training epochs and the memory quantity in Ped2.

and the difference is caused by *StyleConv* and *MappingNetwork*. Besides, multiple sampling is required, which is involved in the decoding process in training phase but omitted in inference phase.

## C.4   MORE DISCUSSION

**Explanation about modulation constraint.**   Ideally, SmoReg is sufficient to constrain the diversity representation changing smoothly, the cooperating modulation parameters should be compact consequently whose effect is similar to the role of modulation constraint. However, it is difficult to sample the entire diversity representation space to make SmoReg effective in natural image, so we add the compatible modulation constraints. In some special cases, OOD images may be included in the latent interpolation path between relatively diverse ID projections, even if the changing is smooth. For example, class "Shirt" in FashionMNIST is sufficiently diverse, which may contain other classes belonging to apparently similar tops (without modulation constraints: $87.7\% \rightarrow 84.4\%$).

**Explanation about sampling-based reconstruction.**   We explain the behavior of sampling-based SmoReg in two main aspects. Firstly, the optimization of reconstruction objective means neighboring embedding should modulate ID projection within the sampling radius $\tau_{tr}$. If IDs are projected compactly, the corresponding sampled region will overlap each other seriously, and the reconstruction targets in Eq. 8 are ambiguous. This provides enough pressure to drive ID projections far away from each other to alleviate the confusing reconstruction naturally. With the sampling radius increasing in the bounded representation space, uniform distribution of ID projections are optimal distribution to ensure the minimum overlap, which encourages SmoReg to sample more untrained region.

Secondly, the diversity space bounded with proper scale $a$ further extends the local smoothness to global smoothness on the premise that sampling possibility in the entire representation space is guaranteed. That is, the above overlap is necessary to bring about. With the aid of enough sampling to cover untrained region, we can smooth non-semantic modulation for IDs. Even if the sampling region are not exactly concatenate with each other (*i.e.* with more overlapped region), it is just equivalent to force model to learning from noisy data (*i.e.* with limited confusion on the sampling boundary) without detriment to the suppression of OODs.

**Explanation about BN features adaptation.**   The adaptation of BN somewhat increases the discriminative ability based on reconstruction error, which promotes model to discard OOD features. However, reconstruction error alone may be disturbed by factors including background, texture, etc. Therefore, quantization error $\|z_{proto}^{sim} - z_{proto}^{comp}\|_2$ is used to enhance the discriminative ability. To this end, BNSim is introduced to preserve semantic information of OODs so that they distribute far away from prototypes.

**Explanation about FashionMNIST results.** To better understand the results on FashionMNIST, we recommend reviewing how DMAD works and the main limitations. Since fashion items within only one class have limited styles (here we mean the standard shape combined with limited color and texture), DMAD (Liu et al., 2023a) naturally stores all prototypical styles to generate standard reconstruction. Then the pyramid deformation module transforms the standard reconstruction with a deformation field to match the real position, size and relative deformation of inputs. Theoretically, all non-semantic diversity can be characterized as geometrical deformation as long as the prototypical memory is sufficient to cover different styles of the given class. And DMAD should solve one-class problems perfectly in FashionMNIST. As reported in Liu et al. (2023a), DMAD achieves $96.3\%AUC$ with grid search. The main reason of performance drop is manually created pseudo OODs are highly associated with geometrical diversity, making DMAD tends to use geometrical measure without reconstruction error and feature distance for scoring function. In that case, BiRDM has $96.6\%AUC$ and still outperforms DMAD, demonstrating BiRDM can model *diversity-agnostic* non-semantic ID without *diversity-specific geometrical inductive bias*.

### C.5 VISUALIZATION

**Disentanglement in Figure 3.** As sampling-based SmoReg may be a feasible way to promote feature disentanglement, we use 3dShapes dataset (Kim & Mnih, 2018) to qualitative show this ability. BiRDM is trained with pixel-level reconstruction loss and the *one-factor-changed* sampling is performed in diversity representation space for the reconstruction results. As shown in Fig. 3, all continuous non-semantic factors are disentangled and the interpolation between gaps of generative factors can also be reconstructed. Since there is no untrained region to forming warped distribution, smoothness forces an efficient representation in bounded diversity space and approximately disentangled linear representation is a solution.

**Non-semantic bias.** We visualize the samples of $Top$-100 smallest OOD scores on CIFAR10 "bird" to qualitatively show some preferences (bias) of the compared reconstruction-based methods. As shown in Figs. 12 and 13, RD and Tilted (Floto et al., 2023) seem to learn to identify OOD based on low-level statistical similarity (*i.e.* the spuriously associated background). Although RD++ in Fig. 14 try to align in-batch foreground, the complex distribution in natural images is hard to be estimated correctly by the MultiProjectionLayer, and the optimal transmission optimization may cause difficulties for the learning discriminative features. This issue can be alleviated by explicitly introducing memory modules to compress prototypical features (van den Oord et al., 2017) as shown in Fig. 15. However, due to the lack of diversity modeling, VQ may miss non-typical diverse ID. Predictably, DMAD prefers geometrical cues to identify OOD, which make some OOD with "average shape" have a low score (*e.g.* a lying cat and the back of a deer may be similar to the body of a bird / ostrich in Fig. 16). Besides, geometrical deformation cannot represent many other non-semantic diversity which makes DMAD fail in natural images. In contrast, BiRDM in Fig. 17 includes minimum background bias and encodes birds with similar shape, pose and species into a more compact distribution, implying that small spatial transformation in latent space corresponds to smoothly changing appearance.

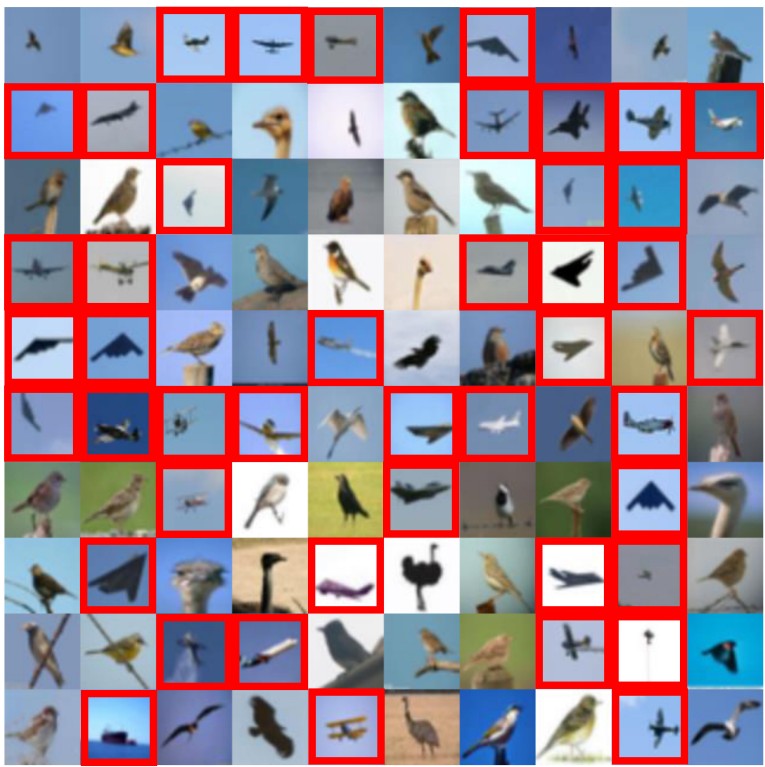

Figure 12: Samples of $Top$-100 smallest OOD scores by RD on CIFAR10 "bird". 43 false negatives are indicated by red boxes.

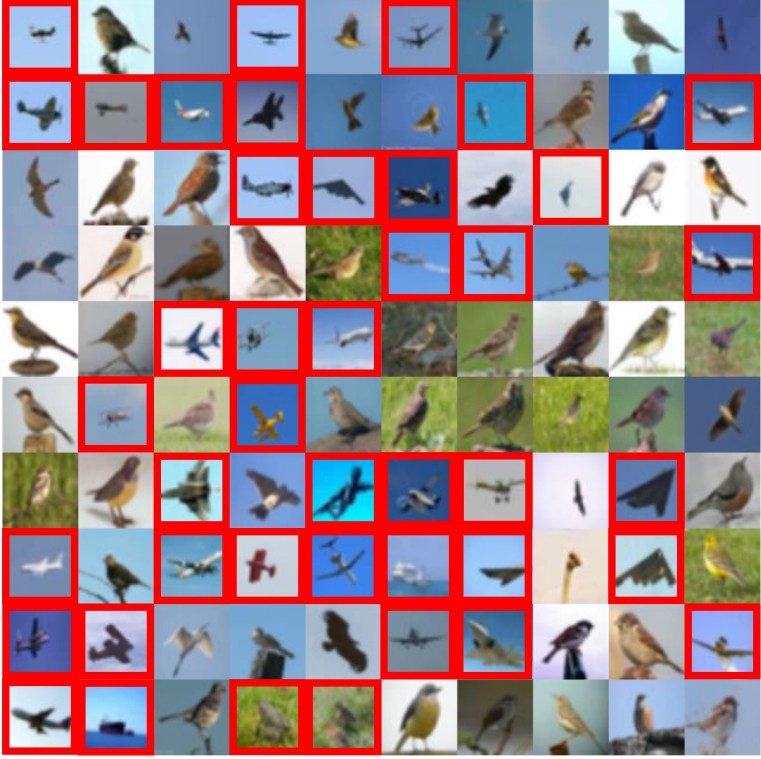

Figure 13: Samples of $Top$-100 smallest OOD scores by Tilted on CIFAR10 "bird". 42 false negatives are indicated by red boxes.

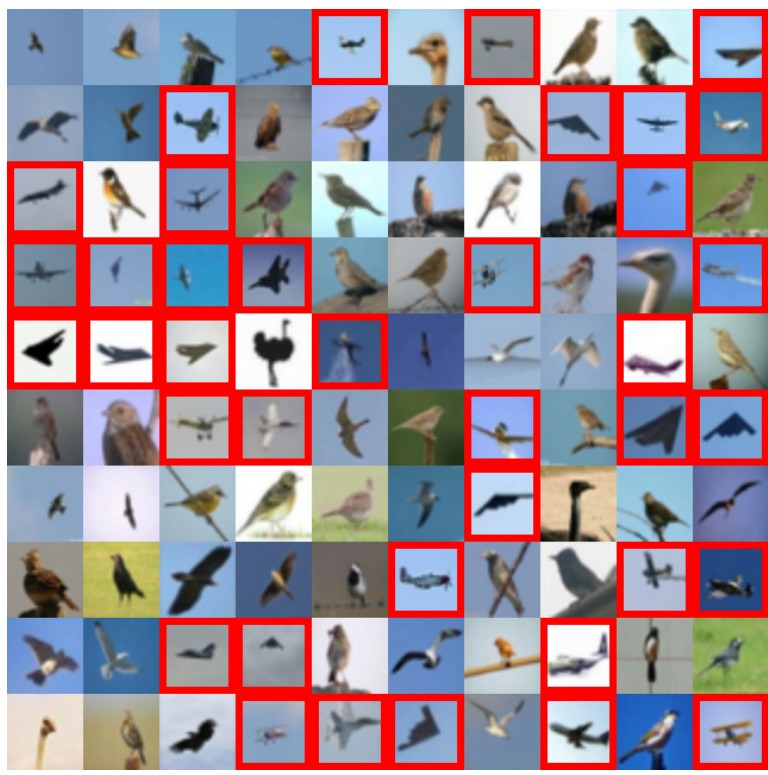

Figure 14: Samples of $Top$-100 smallest OOD scores by RD++ on CIFAR10 "bird". 39 false negatives are indicated by red boxes.

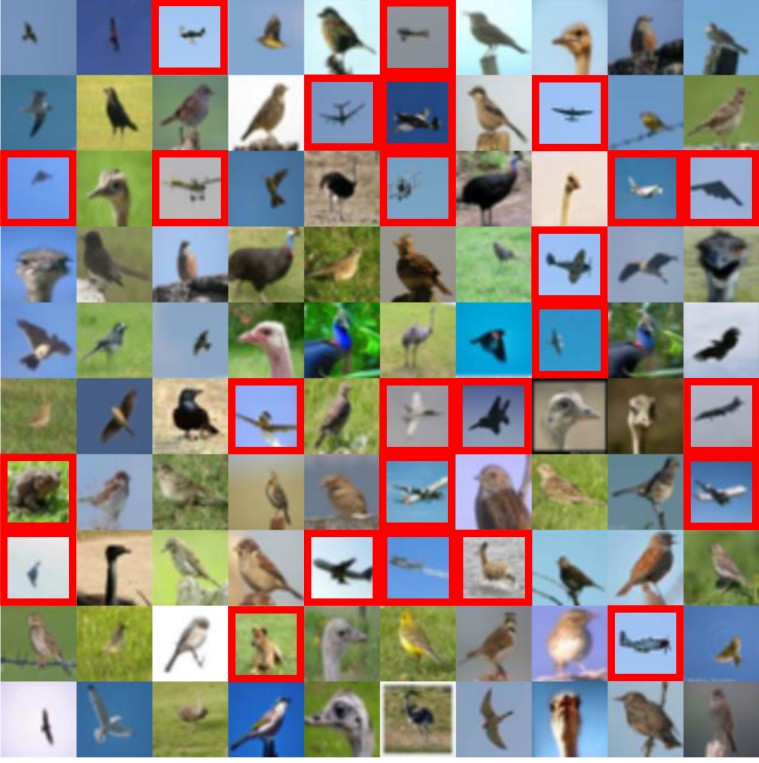

Figure 15: Samples of $Top$-100 smallest OOD scores by VQ on CIFAR10 "bird". 25 false negatives are indicated by red boxes.

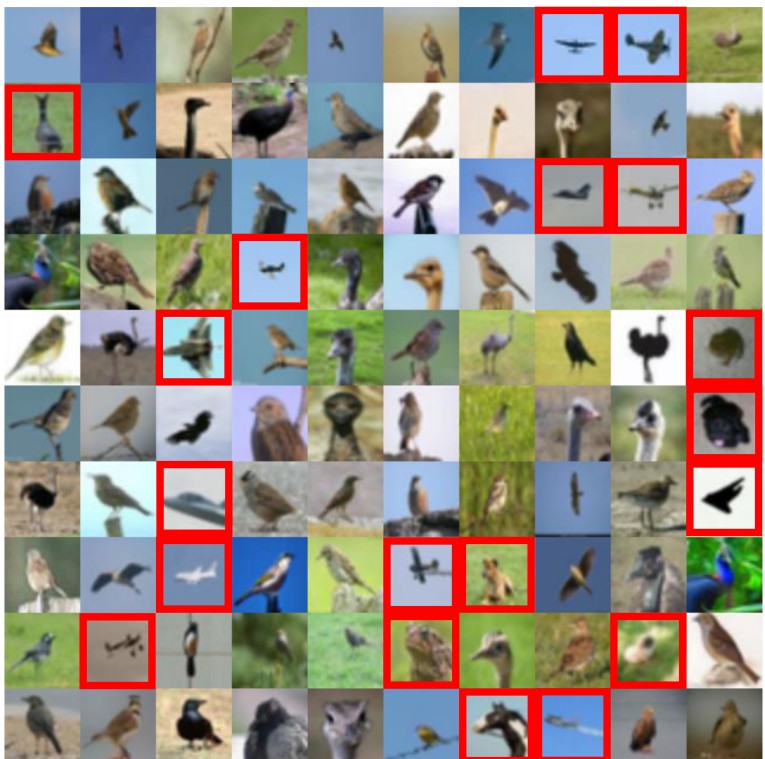

Figure 16: Samples of $Top$-100 smallest OOD scores by DMAD on CIFAR10 "bird". 19 false negatives are indicated by red boxes.

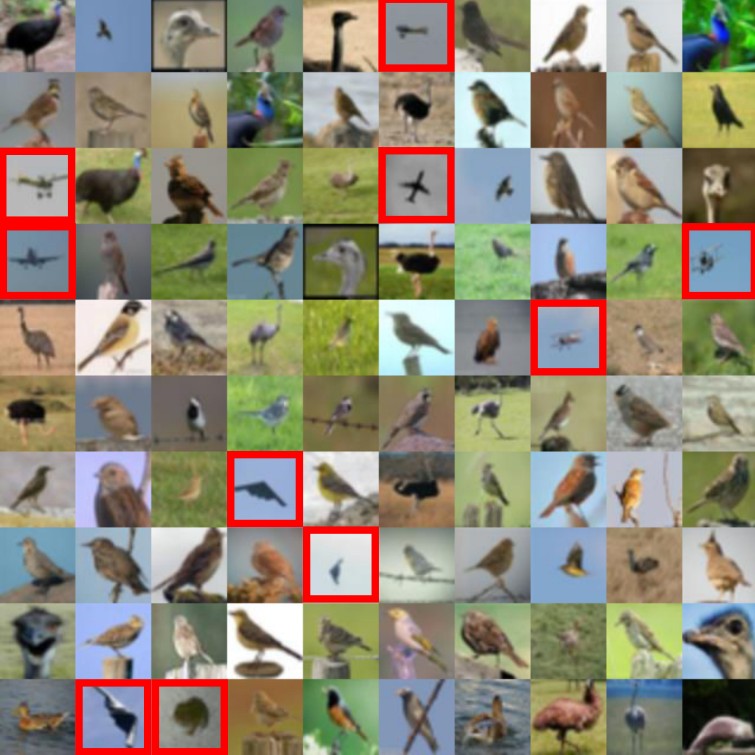

Figure 17: Samples of $Top$-100 smallest OOD scores by BiRDM on CIFAR10 "bird". 10 false negatives are indicated by red boxes.

