# OpenReview forum: "Diversity Modeling for Semantic Shift Detection"
_ICLR.cc/2024/Conference — Submitted to ICLR 2024_

### Official Review · Reviewer_d4iz · 2023-10-30

**Soundness:** 3 good
**Presentation:** 2 fair
**Contribution:** 3 good
**Rating:** 6
**Confidence:** 3

**Summary:**

The paper proposes a set of methods for semantic shift detection. A bidirectional module is proposed to model feature diversity and a regularization technique is proposed for undesired generalizability. Extensive experiments were provided to validate the performance of the method.

**Strengths:**

1. The motivation of the paper is well-grounded: Doing semantic shift detection via exploring non-semantic features. The relationships with previous works are also clarified properly.
2. Experiments demonstrate the effectiveness of the proposed method, outperforming other compared methods in a great margin.

**Weaknesses:**

Though the performance of the proposed framework is superior, I have the following concerns:
1. As illustrated in Figure 2, spatial transformation is done for diversity learning. I wonder how is this different from common data augmentation methods (rotate, flip, clip etc.). Also, how would the proposed BiRDM compare with directly adopting more complex data augmentation methods such as Mixup? Since data augmentation can also be seen as a way for preserving semantic information.
2. The framework is rather complex and requires much hyper-parameter tuning. Thus I have doubt on the applicability of the framework.
3. Figure 1 shows that multiple intermediate features are used. Also BNSim requires auxiliary data. Does these make the proposed framework fair to compare with other baselines? In my view, the type and number of intermediate features should be aligned. Auxiliary data can be added, but more details should be presented (number, type, construction procedure etc.).

**Questions:**

See Weaknesses.

**Details Of Ethics Concerns:**

None.

---

> ### Author Response · Authors · 2023-11-18
> **Thank you for the comments.**
>
> 1. **Spatial transformation vs. common data augmentation:** We assume you are referring to replace spatial transformation with image space augmentation. The problems are as follow:
>    1. It is difficult to figure out how explicit data augmentations can be implemented, since most of them are non-differentiable and difficult to train end-to-end. This causes another issue that the model cannot be diversity-agnostic.
>    2. Limited hand-crafted augmentation cannot span a representation space covering all ID diversity. For IDs, this means most diversity beyond rotating or flipping cannot be reconstructed well. For OODs, this means there are still a lot of untrained regions in the representation space which may fail to suppress OODs.
>    3. We cannot assume that the augmented results still stay on ID manifold, *e.g.* image after MixUp is not a real ID image any more and may becomes OOD in the interpolation path (see **[1]** for more details).
>    4. Actually, **Fig. 2** does not mean that BiRDM can be regarded as a kind of simple data augmentation like rotation. It captures diversity-agnostic non-semantic factors of natural images restrictly beyond rotation, flipping, clipping, etc. and restores them with prototypical ID features.
> 2. **Hyper-parameter:** We report the main ablation of hyperparameters in **Tab. 2**. Other hyperparameters are (1) set to 1, *e.g.* regularization weight except the second term in **Eq. 11**; (2) set to common values, *e.g.* $\beta$ = 0.25 for all VQVAE-related method; (3) selected by pseudo validation set with rotation/blurring as OODs, *e.g.* $\alpha_{1}$...$\alpha_{3}$ in **Eq. 12**. Besides, the weight $\gamma$ of $L_{mod}$ in **Eq. 11** is **not** sensitive as shown in the table below:
>
> | $\gamma$ | CIFAR10-OC | FMNIST-OC |
> | :------: | :--------: | :-------: |
> |   0.1    |    90.4    |   96.4    |
> |   0.25   |    90.9    |   96.3    |
> |   0.5    |    91.0    |   96.4    |
> |   0.75   |    91.2    |   96.3    |
> |    1     |    91.3    |   96.2    |
> | baseline |    86.0    |   95.4    |
>
> 3. **Fairness**
>    1. **Fairness of using intermediate features:** All reconstruction-based methods (the second group in table: RD, RD++, Tilted, VQ, DMAD, BiRDM) are based on the reverse distillation framework and use multiple intermediate features.
>    2. **Fairness of using auxiliary data:** For compared reconstruction-based methods, using auxiliary data will lead to serious **performance drop**.  Note that only BNSim (cooperating with SmoReg) benefits from auxiliary data as we reconstruct both IDs/OODs in a **unified** framework to push OODs away from prototypes (see **Sec. 3.4 Remark** for details).
>    3. **Implementation about auxiliary data:**
>       - **Number:** 1 batch of auxiliary data is optimized after every 5 batches of training data.
>       - **Type:** We use Tiny ImageNet and CIFAR100 as auxiliary data (we use Tiny ImageNet if OOD dataset is CIFAR100 to ensure fairness).
>       - **Construction procedure:** In brief, auxiliary features $z_{aux}^{demod}$ are adapted by *BNSim* to $z_{aux}^{sim}$, and then we reconstruct it as raw inputs $f_{aux}$. As shown in **Fig. 4**, *ConvBN* and *Memory module* are omitted during the reconstructing process of auxiliary data. See **App. A Detailed implementation of BNSim** for more details.
>
> [1] Better mixing via deep representations, ICML 2013.

---

> > ### Comment · Reviewer_d4iz · 2023-11-18
> > **Thank you for the response**
> >
> > The authors have fully addressed my concerns and fixed my misunderstandings of the paper. After going through the other reviews, I appreciate the works the authors have done and their efforts in revising the paper. I am CURRENTLY raising my score.
> >
> > However, I have some further suggestions. Given the "randomness" in the reviewing & rebuttal process, most reviewers are not extremely familiar with the technical details. Going too deep into the details may have negative effects.
> >
> > P.S. For weakness 3 and response 3.1, I am referring to whether the number and types of intermediate features match. For example, every method uses the same blocks' output.

---

> > > ### Author Response · Authors · 2023-11-19
> > > **Thanks for your prompt and positive response.**
> > >
> > > Thanks for your prompt and positive response.
> > >
> > > For the same experiment, every method uses the same number and type of blocks' outputs. For different experiments, features could be selected from ResNet layers1-3 or layers 2-3.

---

### Official Review · Reviewer_oBBb · 2023-10-31

**Soundness:** 3 good
**Presentation:** 3 good
**Contribution:** 4 excellent
**Rating:** 6
**Confidence:** 4

**Summary:**

The paper proposes a method called Bi-directional Regularized Diversity Modulation (BiRDM) for modeling restricted feature diversity in semantic shift detection. The method modulates feature diversity using learnable dynamic modulation parameters in latent space and introduces smoothness regularization and Batch Normalization Simulation (BNSim) to enhance discriminability and separate different semantic distributions. Experimental results demonstrate the effectiveness of the proposed method.

**Strengths:**

- The paper addresses the challenge of modeling non-semantic feature diversity in semantic shift detection.
- The proposed method, BiRDM, introduces novel techniques such as dynamic modulation parameters, smoothness regularization, and BNSim to effectively model diversity-agnostic non-semantic patterns.
- The experimental results demonstrate the effectiveness of the proposed method.

**Weaknesses:**

- The paper lacks a clear motivation for the problem of semantic shift detection and the importance of modeling diversity-agnostic non-semantic patterns.
- The paper could provide more details about the experimental setup, such as the hyperparameters used.

**Questions:**

- Can the proposed method handle different types of semantic shifts, or is it limited to specific patterns?
- How sensitive is the performance of the proposed method to the choice of hyperparameters, such as the regularization weight and the number of modulation stages?

---

> ### Author Response · Authors · 2023-11-18
> **Thank you for the comments.**
>
> 1. **Motivation**
>
>    1. **Semantic shift detection:** Users hope DNNs to reliably reject predictions when facing data beyond its training distribution, rather than providing incorrect results with high confidence. Semantic shift detection is similar to common OOD detection, except that we neither explicitly emphasize ID classification nor use class labels. We believe this setting is general and practical in real-world scenarios thus worth studying.
>    2. **Modeling diversity-agnostic non-semantic patterns:**
>       - **Modeling non-semantic patterns:** As mentioned in **Sec. 3.1**, memory models are used to prevent reconstruction-based methods from reconstructing OODs. But discrete prototypes may discard non-semantic patterns and suppress diverse IDs simultaneously. The solution is to restore non-semantic diversity of prototypes in a well-constrained way. That is, modeling non-semantic patterns aims to solve the trade-off dilemma between reconstructing diverse IDs and suppressing undesired OODs.
>       - **Diversity-agnostic:** Existing methods (*e.g.* DMAD) leverage strong diversity priors to deal with specific non-semantic patterns. They are not general enough for natural images and cause worse performance in semantic shift detection. Therefore, we need to model more general and regularized diversity without prior assumptions about the type of diversity patterns.
>
> 2. **Different types of shifts:** We assume *``different types of semantic shifts''* refers to non-semantic attribute/diversity (because semantic shifts could only be explained as the changes of semantic label which has no different types). If we understand correctly, BiRDM could implicitly handle all different types of non-semantic shifts. An qualitative example in **Fig. 3** shows BiRDM handles view, size, and 3 colors changes without specifying a specific pattern in advance.
>
> 3. **Hyper-parameter & sensitiveness:**
>
>    1. We report the main hyperparameters tuning in **Tab. 2**. Other hyperparameters are (1) set to 1, *e.g.* regularization weight except the second term in **Eq. 11**; (2) set to common values, *e.g.* $\beta$ = 0.25 for all VQVAE-related method; (3) selected by pseudo validation set with rotation/blurring as OODs, *e.g.* $\alpha_{1}$...$\alpha_{3}$ in **Eq. 12**.
>
>    2. **Number of modulation stages:** We test our model on dSprite and find the results is **not** sensitive to modulation stages as long as it is not too limited.
>
>       | $\sharp$ Stages | AUC(%) |
>       | :---------: | :----: |
>       |      0      |  57.6  |
>       |      1      |  97.5  |
>       |      2      |  99.0  |
>       |      3      |  99.9  |
>       |      4      |  100.  |
>
>    3. **Regularization weights:** We set most regularization weights to 1, because they are not adversarial to the reconstruction objective. The only exception is the weight of $L_{mod}$, but it is not sensitive as shown in the table below:
>
>       | $\gamma$ | CIFAR10-OC | FMNIST-OC |
>       | :------: | :--------: | :-------: |
>       |   0.1    |    90.4    |   96.4    |
>       |   0.25   |    90.9    |   96.3    |
>       |   0.5    |    91.0    |   96.4    |
>       |   0.75   |    91.2    |   96.3    |
>       |    1     |    91.3    |   96.2    |
>       | baseline |    86.0    |   95.4    |

---

> > ### Comment · Reviewer_oBBb · 2023-11-22
> >
> > Thank you for sharing the elaboration on semantic shift categories and the ablation study on the regularization weight hyperparameter. This has resolved my confusion, and I will maintain my rating.

---

> ### Author Response · Authors · 2023-11-22
>
> Dear Reviewer oBBb,
>
> As the rebuttal period is ending soon, we kindly wonder if our responses answer your questions and address your concerns. Please let us know if there are any further questions.

---

### Official Review · Reviewer_CShN · 2023-11-01

**Soundness:** 2 fair
**Presentation:** 3 good
**Contribution:** 2 fair
**Rating:** 5
**Confidence:** 4

**Summary:**

To better detect semantic transfer based on reconstruction, this article proposes DMSSD, which mainly uses the newly proposed BiRDM in the architecture to perform modulation feature diversity. Also, modulation constraint, smoothness regularization, and BN simulation coordinating with auxiliary data have been proposed to detect semantic shifts. Experimental results demonstrate the effectiveness of this method.

**Strengths:**

+ The authors present a very detailed explanation of the modeling framework and a good description of the differences and comparisons between the present results.

+ The authors are more detailed in explaining the results of the experiments and are more complete about the experimental parameter settings making the results more convincing.

**Weaknesses:**

1. Contributions and novelties: The innovation of this article is limited.
The proposed architecture is based on previous work ( Diversity-Measurable Anomaly Detection [1] ) and it is modified slightly with demodulation part, and both the framework design and the experimental procedure and phenomenon explanation heavily refer to the work in DMAD, rather than using more of their own The architecture is based on previous work and it is modified slightly with demodulation part.
2. Discussions with previous work: The authors present less discussion of OOD-related work and fewer descriptions of the underlying theories and fundamental methods within the OOD field, making it difficult to highlight the contributions of their work.
3. Presentation issues: The authors had put many key elements of the article, such as Detailed implementation of BNSim and Large-scale dataset, in the appendix at the end of the article, which makes the article much less easy to read and more difficult to understand completely.
4. Experimental results. The experimental datasets, i.e., CIFAR 10 and FashionMNIST are somehow on a small scale and the performance seems to be saturated, taking Tab.2 B as an example. I suggest the authors conduct on some popular datasets including ImageNet. I understand there might be a computational issue, and the authors could choose other subsets of these real-world datasets. This would make the overall paper more convincing.


[1] Wenrui Liu, Hong Chang, Bingpeng Ma, Shiguang Shan, and Xilin Chen. Diversity-measurable anomaly detection. arXiv preprint arXiv:2303.05047, 2023.

**Questions:**

Please refer to the weakness part, my major concern still lies in the major novelty and experimental presentations. The authors could major response in these two aspects.

---

> ### Author Response · Authors · 2023-11-11
>
> Thank you very much for your comments. Due to time limit, we would like to seek confirmation of your suggestion on experiment before formally replying all your concerns. Since **App. C.3** provides experimental results on ImageNet subset (**Weaknesses 3**), we'd like to know more explicitly what kind of additional experiments you're interested in (**Weaknesses 4**)?

---

> > ### Comment · Reviewer_CShN · 2023-11-13
> >
> > The reviewer suggests that the authors should still do experiments on larger and more complete data sets if time is sufficient. The subset of ImageNet is still relatively small, but it solves some of the problems. The reviewer suggests that the author prioritize addressing my other concerns as well as other reviewers' questions.
> >
> > Given the authors' computational resources and the promise of future experiments on large data sets, I am not strict for this weakness.

---

> > ### Comment · Reviewer_CShN · 2023-12-05
> > **Response to the authors' feedback**
> >
> > Thanks a lot for the authors' efforts and hard work.
> > The experimental results added in this manuscript solved my major concerns. Considering I still have concerns about the illustration of key ideas, I tend to maintain my borderline scores and would not be upset if it is accepted. Besides, I do agree the proposed method shows a good improvement compared to DMAD.
> >
> > I suggest the authors move the newly added experimental results to the main manuscript and conduct ablations on large datasets. Results in the main manuscript and unclear presentations may confuse us reviewers.
> >
> > In my view, this proposed method shows promising techniques and good results compared to the prevailing literature but the presentations make this paper hard to understand and follow. I think this paper should be further revised including the figures and main motivations.

---

> ### Author Response · Authors · 2023-11-19
> **Thank you for the comments.**
>
> 1. **Novelties:** ***The comments about lack of novelties are somewhat hasty. We hope to draw your attention to the overlooked MAIN CONTRIBUTIONS (NOT ONLY ARCHITECTURE) and clear up the misunderstood points.***
>
>    - The main contribution lies in ***GENERAL* diversity modeling *WITHOUT* prior assumptions about the type of diversity patterns**, which is necessary for semantic shift detection. In contrast, DMAD relies on structure-guided geometric-specific inductive bias thus can only deal with geometrical anomaly and cause **performance drop** in natural images.
>    - BiRDM is NOT just a slight modification from DMAD. It has  ***COMPLETELY DIFFERENT*** **GENERALITY** (general diversity vs. geometrical diversity), **METHOD** (smooth diversity representation and BN simulation vs. strong assumption about the positive correlation), **IMPLEMENTATION** (diversity-agnostic modeling via SmoReg and BNSim  vs. structure-guided inductive bias), and **PHENOMENON EXPLANATION** (about smoothness, how to shape the latent space, the behavior of BN, disentanglement, etc.). These points are discussed in both main paper and appendix.
>    - Check **Sec.1, 3, App. B (in BiRDM), as well as last two paragraph in Sec. 3.1, first paragraph in Sec. 3.3, Limitation, App. E (in DMAD)** for detailed reasons about how BiRDM is completely different from DMAD.
>
> 2. **Previous work:** We have discussed related works and fundamental methods in **Sec. 1, 2, 3.1 and appendix** (including the difference to DMAD). We briefly summarize the main background here:
>
>    - **Fundamental methods:** Unsupervised reconstruction-based methods are trained on IDs to reconstruct IDs with high quality and potential OODs with low quality.
>    - **Trade-off about reconstruction quality:** Memory models are necessary to prevent reconstructing OODs, since all prototypes are belong to ID. But discrete memory discards non-semantic patterns and suppress diverse IDs together.
>    - ***DMAD and the KEY ISSUE:*** The solution is to restore non-semantic diversity of prototypes in a well-constrained way. However, **DMAD rely heavily on the strong assumption that geometrical diversity is positively correlated with the severity of distribution shift and DMAD cannot restore other diversity in open-world semantic shift detection.** In contrast, our BiRDM is general in modeling **diversity-agnostic** non-semantic patterns to facilitate semantic shift detection.
>
> 3. **Presentation:** We apologize for putting some implementation details and more experiments in the appendix due to space limit. The main consideration is to help readers to focus on the high-level motivations and methods before checking the implementation details. In the camera-ready version, we will include some key details in the experiments section.
>
> 4. **Experiments:** As you pointed out in **W.3**, we do have provided experimental results on ImageNet subset in **App. C.3**. And we add **more experiments on ImageNet** as you suggested in **W.4**. Besides, CIFAR10 is not simpler than large-scale ImageNet dataset according to [1], and the performance does not saturate as well.
>
>    |IN30|CUB|Dogs|Pets|Flowers|Food|Places|Caltech|DTD|
>    |:-:|:-:|:-:|:-:|:-:|:-:|:-:|:-:|:-:|
>    |RD|62.6|67.2|68.0|87.4|65.4|64.3|54.6|81.5|
>    |RD++|57.4|70.3|73.0|$\underline{91.0}$|66.6|61.3|50.6|$\underline{85.5}$|
>    |Tilted|66.9|78.8|81.4|90.4|67.8|61.8|58.0|79.6|
>    |VQ|67.0|$\underline{80.3}$|$\underline{81.9}$|87.7|67.9|**66.1**|$\underline{66.4}$|82.5|
>    |DMAD|$\underline{67.7}$|74.2|76.3|87.8|$\underline{79.3}$|63.7|62.2|73.9|
>    |BiRDM|**82.0**|**84.8**|**87.2**|**93.7**|**86.0**|$\underline{65.5}$|**70.1**|**89.9**|
>
>    |IN100|iNaturalist|SUN|Places|DTD|
>    |:-:|:-:|:-:|:-:|:-:|
>    |RD|85.0|60.3|56.8|$\underline{81.9}$|
>    |RD++|83.5|49.9|52.8|80.4|
>    |Tilted|73.4|$\underline{76.1}$|$\underline{72.3}$|76.4|
>    |VQ|60.7|64.5|58.6|76.5|
>    |DMAD|**90.2**|64.9|58.7|81.7|
>    |BiRDM|$\underline{86.8}$|**85.6**|**80.7**|**85.9**|
>
>    [1] OpenOOD: Benchmarking Generalized Out-of-Distribution Detection, NeurIPS 2022.

---

> ### Author Response · Authors · 2023-11-22
>
> Dear Reviewer CShN,
>
> As the rebuttal period is ending soon, we kindly wonder if our responses answer your questions and address your concerns. Please let us know if there are any further questions.

---

### Official Review · Reviewer_k7Jp · 2023-11-02

**Soundness:** 2 fair
**Presentation:** 2 fair
**Contribution:** 2 fair
**Rating:** 3
**Confidence:** 3

**Summary:**

The paper proposes a method for semantic shift detection based on the reconstruction of discretized features. The overall idea is to better model the diversity, or fine-graininess of features. The starting method, used in previous works, is to obtain some feature vectors, then use a memory bank to quantize them, and then reconstruct the initial samples. High reconstruction errors signify OOD samples. The paper suggests some limitations to this framework and suggests solutions.

Problem 1: all samples are hard to reconstruct due to memory quantization. Solution: proposed BiRDM: demodulate features, to make them simpler, less diverse and similar to memory features then modulate them back to gain back the diversity. This makes all samples easier to reconstruct.

Problem 2: there seems to be a problem that fine-grained features are not smooth enough, and they can generate noisy modulation params, thus a smoothness constraint is proposed.

Problem 3: the features after ConvBN cannot discriminate between ID and OOD data. Although this is helpful for bad reconstruction of OOD data. Nevertheless, a model (BNSim) is proposed that can bypass the memory quantization and still discriminate between ID and OOD samples. This BNSim model is trained by reconstruction on auxiliary data. Then, its output can be compared to the memory quantized output, and the difference can be used as part of the semantic shift detection score.

**Strengths:**

- S1. Most proposed solutions are sound and can benefit semantic detection. The demodulation-modulation approach seems appropriate for avoiding the loss of fine-grained features in the quantization process.
- S2. The proposed method achieves good results compared to other baselines.

**Weaknesses:**

- W1. The paper is hard to read and the motivation is unclear many times. More details will follow.

- W2. The method is very complex, with a lot of components where some address the weaknesses but also affect the benefits of other components. The complexity together with the fine balance that the components need means that probably the method is hard to tune.

- W2.2 For example, the memory quantization is used because OOD samples should be harder to quantize thus OOD samples are harder to reconstruct. But BiRDM makes *every* sample easier to reconstruct. There is a fine balance that these two components must achieve, and this could be potentially worrisome as the method could be hard to optimize. I am open to clarification in this regard.

- W3. It is not clear if the compared methods are also using additional data. Is this the case?

- W4. Table 1 mostly presents the results of previous methods in the current setting, as evaluated by the current paper. This can be problematic, as we don’t have a proper comparison to the results as presented in previous work. Hyperparameter tuning and model selection can influence the results to a high degree. The same amount of compute should be invested into hyperparameter tunning/model selection of all methods. Was this the case?

- W5. There are some missing details: such kind of data is used as auxiliary data, what is the exact architecture of the multiple modulation stages, and what is the connection to reverse distillation (Deng & Li, 2022)?

- W6. There are some unclear statements and motivations, as follows:

- W6.1 Referring to smoothness constraint: “In this way, potentially high (low) reconstruction error for IDs (OODs) can be alleviated.” It makes sense that the smoothness constraint will alleviate high reconstruction errors for *all* samples. It is not clear why it can alleviate high errors only for ID samples but not for OOD samples. Could the authors explain this?

- W6.2 “SmoReg firstly projects diversity feature $z_{diver}$ to a D-dimension bounded diversity representation space by orthogonal matrix P to capture more compact non-semantic factors”. How do we know that this space captures non-semantic factors but it does not capture semantic factors? Is there any constraint to this effect?

- W6.3 “minimizing $L_{smo}$ with aid of enough sampled \$tilde{z}_{proj}$ (as explained in App. C) ensures smooth non-semantic diversity modeling”. Again, what makes this reflect only the non-semantic diversity, but not the semantic diversity?

- W6.4 “SmoReg targets at smoothing the unoccupied regions in diversity representation space to suppress semantic shifts without affecting the real reconstruction of IDs”. What constraint makes it suppress semantic shifts? Seems like the smoothness constraint would help all kind of diversity (semantic or not) to be better kept in the features. It is not clear how to act differently on semantic vs non-semantic features.

- W6.5 “BN may incorrectly adapt OODs to training distribution with global affine transformation and decrease the feature discriminability”. Why is this a problem, if the BN adapts OOD samples to ID distribution, then the reconstruction error should be high, thus it would be easy to detect the OOD samples.


- W6.6 The difference $z_{proto}^{sim} - z_{proto}^{comp}$ between BNSim features ($z_{proto}^{sim}$) and memory quantization features ($z_{proto}^{comp}$) is used as OOD detection score. This assumes that ID samples do not have high quantization error while OOD samples have high quantization error when compared to BNSim features. There seems to be a contradiction here: The authors note that the convBN cannot discriminate between ID and OOD samples but the proposed score only works if the convBN features are quantized differently (such that ID samples result in quantized features with low error compared to BNSim features, and OOD samples result in quantized features with high error compared to BNSim features). Can the authors explain this apparent contradiction?

**Questions:**

Can the authors clarify the points raised in the Weaknesses section?

**Details Of Ethics Concerns:**

Can the authors give more details about the visualization in Figure 6?

---

> ### Author Response · Authors · 2023-11-18
> **Thank you for your careful and valuable reviews.**
>
> **W2-2.2&6.1-6.4** Alleviating the reconstruction-suppression trade-off is what BiRDM tries to do. Different components(memory, smooth diversity modeling, BN simulation) **do not contradict** to each other.
>
> 1. Since prototype learning is a relatively independent process constrained by VQ loss before modulation, the memory representation ability is affected by the number of prototypes but not diversity modeling.
> 2. BiRDM makes diversity representation capacity large enough to **restore the residual ID diversity** removed by memory, while **OODs are much more difficult to be reconstructed** by imposing SmoReg. The two targets are actually **consistent** with the smoothness regularized diversity modeling.
>    - **ID reconstruction:** smoothly changing diversity modulation encourages ID projections to uniformly distribute in diversity space with low reconstruction error.
>    - **OOD suppression:** Since OODs are also projected and truncated to this bounded, smooth and dense(via enough sampling $\tilde{z}_{diver}^{proj}$) diversity space,  the projections will be close to IDs with high reconstruction error.
>    - From the view of information theory, for OODs, information preserved in prototype quantization is $I(z_{OOD};Z_{ID}^{proto})$, and information preserved in diversity space after demodulation is $I(z_{OOD};Z_{ID}|Z_{ID}^{proto})$. Thus, a lot of OOD information has been lost:$H(z_{OOD}) - I(z_{OOD};Z_{ID}|Z_{ID}^{proto})- I(z_{OOD};Z_{ID}^{proto})>0$,causing high reconstruction error.
> 3. Non-semantic factors refer to ID diversity (e.g. style variations) and semantic factors refer to different clusters with semantic shift (e.g. different classes).
> 4. The only constraint contradictive to reconstruction target is $L_{mod}$ which constrains the modulation amplitude, but we show $\gamma$ is **not** sensitive:
>
> | $\gamma$ | CIFAR10-OC | FMNIST-OC |
> | :------: | :--------: | :-------: |
> |   0.1    |    90.4    |   96.4    |
> |   0.25   |    90.9    |   96.3    |
> |   0.5    |    91.0    |   96.4    |
> |   0.75   |    91.2    |   96.3    |
> |    1     |    91.3    |   96.2    |
> | baseline |    86.0    |   95.4    |
>
> **W3** For compared methods, reconstructing auxiliary data leads to serious **performance drop** due to the undesired generalization. And we use auxiliary data in totally different way from previous works as mentioned in **Sec. 3.4 Remark**: 1. labels are not required for excluding IDs in auxiliary data; 2. we optimize both IDs and OODs with **unified** reconstruction objective to preserve enough semantic information.
>
> **W4** For compared methods, we search their hyperparameters on testing set (to save tunning skills), so their performance we reported should be higher than the reality. For our method, hyperparameters are: 1. set to 1, e.g. regularization weight except $L_{mod}$(not sensitive as shown above); 2. set to common values, e.g. $\beta$=0.25 is widely used for VQ-related method; 3. selected on pseudo validation set with rotation/blurring as OOD, e.g. $\alpha$ in Eq. 12. Hyperparameter tuning cost is comparable.
>
> **W5** We use TinyImageNet/CIFAR100 as auxiliary data(use TinyImageNet if OOD is CIFAR100 to ensure fairness); The modulation layer($M_{1}…M_{3}$)  exactly matches StyleConv in StyleGAN2(receives an input from previous decoder layer and a style vector from corresponding diversity mapping);  BiRDM uses the same decoder($D_{1}…D_{3}$) as Reverse Distillation(RD) to reconstruct encoded features, i.e. the architecture of our decoder layer is related to RD.
>
> **W6.5** BN somewhat increases the discriminative ability based on reconstruction error. However, reconstruction error alone may be disturbed by factors including background and texture. Therefore, quantization error is used to enhance the discriminative ability. To this end, BNSim is introduced to preserve semantic information of OODs so that they distribute far away from prototypes. We will clear up this point in our paper.
>
> **W6.6** We assume the question is: *Since BNSim tracks ConvBN and ConvBN cannot discriminate between ID and OOD, why is the quantization error from BNSim features discriminative?*
>
> 1. BNSim tracks ConvBN features **only for training IDs**.  For auxiliary data including potential OODs, their BNSim features are different from ConvBN features and can be discriminative.
> 2. BNSim performs warped transformation in contrast to global affine transformation. By reconstructing auxiliary data without memory quantization, BNSim features of OODs tend to distribute far from prototypes with high quantization error.
>
> **EC.1 We revise explanation and illustration about Fig. 6 in the updated submission App. C3. Data split:** 1. Training data is composed of ellipses located away from the centric region; 2. Auxiliary data contains hearts at all locations; 3. Testing data contains ellipses at the centric region(ID), and squares(OOD) at all locations;**Model:** BiRDM without SmoReg & BNSim; BiRDM with SmoReg only; full-components BiRDM.

---

> ### Author Response · Authors · 2023-11-22
>
> Dear Reviewer k7Jp,
>
> As the rebuttal period is ending soon, we kindly wonder if our responses answer your questions and address your concerns. Please let us know if there are any further questions.

---

### Official Review · Reviewer_BvVD · 2023-11-07

**Soundness:** 3 good
**Presentation:** 2 fair
**Contribution:** 2 fair
**Rating:** 3
**Confidence:** 4

**Summary:**

This paper target at out of distribution detection, specifically focus on the semantic shift. It claims that there is a trade-off between non-semantic in-distribution diversity and over generalization to out-of-distribution semantic shift. The paper lies in the scope of reconstruction-based error detection, and adopts a method that disentangle the representations of non-semantic diversity and the semantic prototypes from input features. Benefit from the non-semantic diversity modeling, the proposed method learns compact prototypical features. Authors conduct experiments on two benchmarks, with suppressing results to existing works.

**Strengths:**

(1)	The authors proposed a new technique, that is to demodulate the original input features to remove non-semantic diversity, and thus the model can learn compact semantic prototypes which is helpful for detecting out-of-distribution semantic shift. The demodulation is combined with a modulation layer to better reconstruct the input. The overall technique make sense.
(2)	The experiments on CIFAR10 show good improvements to existing methods.

**Weaknesses:**

(1) The presentation needs improvment. There are some unsupported claims. For example, the author claims that there is a trade-off between non-semantic in-distribution diversity and over generalization to out-of-distribution semantic shift, however, as suggested in DMAD[3], the in-distribution diversity is mainly related to covariate shift. Intuitively, to better detect the OOD semantic shift is to learn compact and discriminative prototypical features. The claimed motivation should be doubted according to this.
(2) The experiments are not convincing, too. The paper only shows improvements on CIFAR10, and on FashionMNIST, authors claimed that it only contains image-level geometrical diversity which is also not supported (i.e., how to measure the claimed “image-level” or “feature-level” diversity of a dataset?). The benchmarks used in [1] and [14] should also be considered. What’s more, the ablation experiments are only conducted on a specific class of CIFAR10, why the class is representative? It is important to show the overall-class performance, especially for Table 2.
[1] OpenOOD: Benchmarking Generalized Out-of-Distribution Detection, NeurIPS 2022.
[2] Diversity-measurable anomaly detection. Arxiv 2023
[3] Diversity-measurable anomaly detection. CVPR 2023

**Questions:**

See the weaknesses.

---

> ### Author Response · Authors · 2023-11-18
>
> ***We figure that the comment is a hasty copy paste from the review of our previous submission. There are full of serious misunderstandings and errors of fact for this paper.***
>
> 1. **Reviewer's intuition about compact prototypes is *WRONG***. We make clarification again as follows:
>
>    1. Diversity modeling is **necessary** for semantic shift detection too. According to the basic assumption of reconstruction-based methods, our target is reconstructing diverse IDs with low reconstruction error and suppressing OOD reconstruction.  **Sec. 3.1, as well as DMAD**, shows that if we learn finite prototypes (even though they are *``compact and discriminative''*) only, diverse IDs beyond prototypes will be reconstructed as poor as OODs due to the limited representation capacity of prototypes and we cannot discriminate them by comparing reconstructed error.
>    2. Due to the lack of real OODs, it is **DIFFICULT** for previous methods capture *discriminative* features, especially between diverse IDs and near-OODs. This is the reason why BiRDM tries to maintain the discriminative features by shaping a smooth non-semantic ID region and pushing potential OODs away.
>    3. Although ID diversity is mainly related to covariate shifts, that **DOES NOT** implies that modeling ID diversity **CANNOT** benefits semantic shift detection. Actually, it can.
>    4. For semantic detection, the geometrical prior used in DMAD is no longer available. We have to solve the trade-off dilemma in **diversity-agnostic** way, which makes BiRDM a completely different method to DMAD.
>
> 2. **Most comments on experiments are not relevant at all.**
>
>    1. > The paper only shows improvements on CIFAR10, ... The benchmarks used in [1] should also be considered
>
>       Actually, We **do have reported** the performance on benchmarks used in [1], including ImageNet (in **App. C** as mentioned in **Sec. 4.2**).
>
>    2. > on FashionMNIST, authors claimed that it only contains image-level geometrical diversity ...
>
>       We **DO NOT** claim like that in this paper. (By the way, we **HAVE** made clarification on this point in previous rebuttal which didn't get any attention.)
>
>    3. > The benchmarks used in [14] should also be considered.
>
>       There is **NO citation [14]** at all. If you are referring to surveillance video and industrial images reported in DMAD (the citation index of DMAD used in our previous manuscript is [14]), researchers in OOD community never test these datasets (AD setting) because they include covariate shifts rather than semantic shifts.
>
>    4. > the ablation experiments are only conducted on a specific class of CIFAR10 ... It is important to show the overall-class performance.
>
>       We **DO have reported** the overall-class performance in our ablation study, as shown in Tab. 2(a).

---

> ### Author Response · Authors · 2023-11-22
>
> Dear Reviewer BvVD,
>
> As the rebuttal period is ending soon, we kindly wonder if our responses answer your questions and address your concerns. Please let us know if there are any further questions.

---

### Author Response · Authors · 2023-11-20
**We thank every reviewers' time and efforts!**

According to the reviews, we are currently re-organizing our manuscript and a new version has been uploaded. The main modifications in this version are:

- **More explanations and discussions**: For example, standard BN increases the discriminative ability based on reconstruction error (adapt OOD features to prototypical distribution), but reconstruction error alone may be disturbed by factors including background, texture, etc. Therefore, quantization error is used to enhance the discriminative ability. To this end, BNSim is introduced to preserve semantic information of OODs so that they distribute far away from prototypical distribution.

- **More experiments on hyperparameters sensitivity**: We show more modulation stages means higher performance and it is not sensitive as long as it is not too limited. Besides, most regularization weights could be set to 1 without tuning and the weight of adversarial constraint $L_{mod}$ is not sensitive.

- **Illustrations and explanations of our toy experiments**: We add more explanations and post an illustration to show the details of our toy experiment in **Fig. 6** qualitatively.

- **More experiments on large-scale datasets**: We add more experiments on large-scale ImageNet. The experimental results basically support our observation in the main paper.

---

### Comment · Area_Chair_51AP · 2023-12-05
**Final Update**

Dear Reviewers,

Please take this chance to carefully read the rebuttal from the authors and make any final changes if necessary.

Please also respond to the authors that you have read their rebuttal, and give feedback whether their rebuttal have addressed your concerns.

Thank you,

AC

---

### Meta-Review · Area_Chair_51AP · 2023-12-15

**Metareview:**

In this paper, the authors address the problem to detect semantic shift by proposing a reconstruction-based error detection. The proposed method disentangles the non-semantic diversities and the semantic shifts, with an overall idea to better model the diversity. The proposed method is based on existing works where one uses memory bank to quantize the feature and then reconstruct the image, where high reconstruction errors indicates OOD. For this work, in particular, the authors propose a bidirectional module to model feature diversity and a regularization technique is to handle undesired generalizability. The authors conduct experiments on multiple benchmarks, with results surpassing existing works.

This is a borderline case with diverging reviews, coupled with certain issues such as copied review from previous venue without changes. The AC has carefully read the paper, its relevant literature and all the reviews/discussions here. On the strength side, the AC considers the technical novelty from this work sufficient and the paper has the potential to be accepted given careful revision. The results on several benchmarks are also promising.

On the weakness side, the AC shares some common concerns with the reviewers, even though some of the reviews do have the issue of being directly copy-pasted from a previous venue.

1) As several reviewers have commonly pointed out, the proposed method is overly complicated and the paper is in general hard to follow. This over complication then leads to additional claims/conclusions/choices not being sufficiently verified by the experiment, making it overall less insightful.

2) Experimental section could be improved significantly. The AC finds it hard to obtain sufficient amount of details even after searching carefully in the supplementary. For instance, the details of the ImageNet100 dataset was never mentioned anywhere, and similarly other information regarding the detailed experimental settings are hard to find.

3) The AC also agrees with the reviewers that the scale of the experiments are too small, despite the argument from author that Cifar10 is hard. Experiments on the complete ImageNet (instead of subset) or following established benchmarks such as OpenOOD would be great.

4) The AC does not fully agree with the authors' justification of using auxiliary data because other methods would perform worse using them. More detailed discussions need to be devoted here.

Given the above weaknesses, the AC considers that significant amount of revisions need to happen and the paper is not ready yet for publication at ICLR.

**Justification For Why Not Higher Score:**

Please kindly refer to the weaknesses in the comment section. This paper has the potential to be accepted at future venues but needs significant improvement in both presentation and experiment.

**Justification For Why Not Lower Score:**

N/A

---

### Decision · Program_Chairs · 2024-01-16

Reject